# Effectiveness of Meditation Techniques in Treating Post-Traumatic Stress Disorder: A Systematic Review and Meta-Analysis

**DOI:** 10.3390/medicina60122050

**Published:** 2024-12-12

**Authors:** David W. Orme-Johnson, Vernon A. Barnes, Brian Rees, Jean Tobin, Kenneth G. Walton

**Affiliations:** 1Department of Psychology, Maharishi International University, Fairfield, IA 52557, USA; 2Georgia Prevention Institute, Augusta University, Augusta, GA 30912, USA; vbarnes@augusta.edu; 3Medical Corps, U.S. Army Reserve, San Luis Obispo, CA 93401, USA; brian.rees54@gmail.com; 4Transcendental Meditation for Women, Maharishi Vedic City, IA 52556, USA; jeantobin108@gmail.com; 5Institute for Prevention Research, Maharishi International University, Fairfield, IA 52557, USA; kwalton@miu.edu

**Keywords:** post-traumatic, PTSD, meditation, mindfulness, transcendental meditation, systematic review, meta-analysis

## Abstract

*Background and Objectives*: Post-traumatic stress disorder (PTSD) is a debilitating condition worldwide. The limited effectiveness of current psychological and pharmacological treatments has motivated studies on meditation techniques. This study is a comprehensive, multiple-treatments meta-analysis comparing the effectiveness of different categories of meditation in treating PTSD. *Methods and Materials*: We followed Prisma guidelines in our published protocol to search major databases and to conduct a meta-analysis of the studies. *Results*: We located 61 studies with 3440 subjects and divided them logically into four treatment groups: Mindfulness-Based Stress Reduction (MBSR, 13 studies); Mindfulness-Based Other techniques (MBO, 16 studies), Transcendental Meditation (TM, 18 studies), and Other Meditations that were neither mindfulness nor TM (OM, 14 studies). Trauma populations included war veterans, war refugees, earthquake and tsunami victims, female survivors of interpersonal violence, clinical nurses, male and female prison inmates, and traumatized students. Of those offered, 86% were willing to try meditation. The baseline characteristics of subjects were similar across meditation categories: mean age = 52.2 years, range 29–75; sample size = 55.4, range 5–249; % males = 65.1%, range 0–100; and maximum study duration = 13.2 weeks, range 1–48. There were no significant differences between treatment categories on strength of research design nor evidence of publication bias. The pooled mean effect sizes in Hedges’s g for the four categories were MBSR = −0.52, MBO = −0.66, OM = −0.63, and TM = −1.13. There were no appreciable differences in the study characteristics of research conducted on different meditations in terms of the types of study populations included, outcome measures, control conditions, gender, or length of time between the intervention and assessment of PTSD. TM’s effect was significantly larger than for each of the other categories, which did not differ from each other. No study reported serious side effects. *Conclusions*: All categories of meditation studied were helpful in mitigating symptoms of PTSD. TM produced clinically significant reductions in PTSD in all trauma groups. We recommend a multisite Phase 3 clinical trial to test TM’s efficacy compared with standard treatment.

## 1. Introduction

**The Problem**. Traumas from any source may stretch an individual’s physical and mental adaptive resources beyond their ability for timely recovery, causing Post-Traumatic Stress Disorder (PTSD) [1]. PTSD symptoms include hyper-arousal (looking out for threats), dissociative reactions (flashbacks) that make one feel like the traumatic event is happening again, nightmares that can cause people to wake up suddenly, sometimes kicking and screaming, and insomnia. Physical signs can include increased heart rate, rapid breathing, muscle tension, pupil dilation, or increased blood pressure. Affected persons may have difficulty concentrating, and PTSD is co-morbid with other conditions such as major depression, anxiety, substance abuse, poor physical health, and suicide. Other symptoms include avoiding reminders: avoiding places or situations that bring back memories of the trauma; changes in behavior: withdrawing from friends and family or keeping busy to avoid talking about the event; loss of interest: losing interest in activities one used to enjoy; and substance misuse: trying to cope with feelings by misusing alcohol or drugs. It is clear that PTSD takes a severe toll on the mental, physical, and financial wellbeing of veterans, civilians, and their families [2]. A national survey found that major depression is 3 to 5 times more likely in those with PTSD [3]. These symptoms of PTSD unfortunately degrade the quality of life of individuals, their families, and communities throughout the world, resulting in suicide, substance abuse, divorce, child and marital neglect and abuse, and community disruption.

**Prevalence**. The World Health Organization (WHO) estimates that 3.9% of the global population experiences PTSD at some point in their life [4]. This means that for the current world population of 8.2 billion, 320 million people, which is approximately the population of the U.S., will suffer from PTSD during their lifetime. PTSD affects approximately 6% to 7% of adults in the U.S. over their lifespan [5]. In the U.S., there are conservatively over 1 million veterans suffering from PTSD today. The total number of those afflicted balloons to over 10 million if civilians are included. The incidence is greater in veterans and active-duty military personnel, estimated to affect 14% of those who were deployed in wars in Afghanistan and Iraq [6]. Estimates of lifetime prevalence of PTSD for military personnel range from 10% to 29% [7]. This is an enormous unsolved public health problem, and it needs to be quickly resolved to bring much needed relief to this huge suffering population.

**Treatment**. Current treatments are typically pharmaceuticals along with one or more forms of Cognitive Behavioral Therapy (CBT). A study of 186,240 VA patients with PTSD found that 80.1% had been prescribed psychiatric medications [8]. Another study found that approximately half (47%) receive psychiatric prescriptions. Pharmaceuticals can be targeted toward symptoms, such as depression, anxiety, insomnia, and nightmares. Particularly, anti-depressants designed to increase the levels of serotonin in the body have been used. Serotonin has a role to play in learning, memory, and happiness, as well as regulating body temperature, sleep, sexual behavior, and hunger. Insufficient serotonin is thought to play a role in depression, anxiety, mania, and other health conditions. Research indicates that the consumption of psychotropic drugs was not adequately monitored [9]. It is generally recognized that there is no drug or drug cocktail that is specific for or can effectively cure PTSD.

The most frequently studied psychotherapies for military-related PTSD are Cognitive Processing Therapy (CPT) and Prolonged Exposure (PE) [10,11]. A number of other studies looking at “real world” effectiveness have shown that many veterans find the “best” treatments intolerably uncomfortable, with completion rates of 10% or less [12,13,14]. Moreover, about half of the veterans with PTSD do not seek treatment. Of the half that do seek treatment, only half of them get “minimally adequate” treatment. In one study, only 10% of patients had at least 12 therapy visits in the 6 months following diagnosis [9]. But even those who receive the best available treatment all too often are still not adequately served. A 35-year review of randomized controlled trials (RCTs) from 1980 to 2015 found that the two CBT treatments regarded as the “gold standard” at that time, i.e., Prolonged Exposure (PE) and Cognitive Processing Therapy (CPT), were inadequately efficacious; two-thirds of veterans with PTSD treated with PE or CPT retained their diagnosis after completing treatment [10,13,14]. Eye Movement Desensitization and Restructuring (EMDR) has also shown consistent reduction of symptoms of PTSD with the completion of 12–16 60 min weekly sessions [11].

The limitations in effectiveness and tolerability of the most frequently employed PTSD treatments all too often leave veterans in the unsatisfactory position of fending for themselves. That is, many patients cannot tolerate reliving their traumatic experiences and simply refuse to participate in trauma-focused therapy. Moreover, patients may be limited by access to a therapist who has been trained in trauma-focused therapy, even if they are willing to undergo it, as well as by cost and/or insurance coverage [11]. There is a need to identify more effective and life-supporting treatments for veterans [10]. These considerations raise “important practical questions, particularly the value of emotionally demanding therapies, such as PE and CPT, relative to comparably efficacious and more tolerable interventions, such as PCT (Present Centered Therapy), Transcendental Meditation, and “sertraline”, a selective serotonin reuptake inhibitor [15]. A variety of essentially non-trauma focused meditation practices have been considered and hold enormous promise in the treatment of PTSD [16,17].

### Meditation in the Treatment of PTSD

The first study to use a meditation technique to treat PTSD (then called post-Vietnam adjustment) was Brooks and Scarano, 1985, conducted at the Veterans Outreach Program in Denver Colorado [18]. It compared TM and psychotherapy over 3 months for Vietnam veterans. The effects were strong for a broad array of outcomes (PTSD symptoms, anxiety, depression, drug and alcohol use, job and family life). The study was small and was not followed up for two decades. Before 1985, there had been 15 years of research on TM, which began in 1970 with Wallace et al.’s publication of TM’s physiological effects in Science [19], the American Journal of Physiology [20], and Scientific American [21], and Orme-Johnson’s study in Psychosomatic Medicine [22]. Wallace found that the physiological changes during TM were in the opposite direction from the “Fight or Flight’” stress response. Orme-Johnson found that TM reduced baseline levels of stress markers during meditation and that it changed the response to stress outside of meditation to fewer multiple stress responses and more rapid recovery from the stressor. By 1985, these studies had been complemented by 350 additional studies on TM showing its metabolic, biochemical, cardiovascular, and electroencephalographic changes, and its benefits for physiological efficiency and stability, cardiovascular and general health, motor and perceptual ability, athletic performance, prison and drug rehabilitation, productivity, and quality of life [23,24]. Today there are over 750 scientific papers on TM [25,26,27,28].

The next major step in the use of meditation as a treatment in psychology and health started in 1979 when Jon Kabat-Zinn began teaching Mindfulness-Based Stress Reduction (MBSR) at the University of Massachusetts Medical School. He created a standardized protocol for teaching mindfulness, MBSR, which facilitated research. This research began in 1982 with Kabat-Zinn’s preliminary study of mindfulness for chronic pain [29] and became a popular concept with the publication of his book in 1990 [30]. Although initially developed for stress management, it has evolved to encompass the treatment of a variety of health-related disorders. Between 1966 and 2021, there were over 16,000 studies published related to mindfulness, with the number of publications increasing by an average of 23.5% per year from 2010 to 2020 [31].

“Mindfulness”, even the manualized MBSR, is a heterogenous approach with many components, not a single practice, and so far there has been few research studies on which of the components are the “active ingredients”. Moreover, a reviewer [32] pointed out that “the currently applied mindfulness-based interventions show large differences in the way mindfulness is conceptualized and practiced… [There is an] absence of consensus about an operational definition of ‘mindfulness’. Finally, it is noteworthy that the word ‘mindfulness’ is frequently used as a construct, a mental state, or as a number of practices designed to achieve this state, raising concerns as to what current studies are actually measuring when they claim they are measuring mindfulness and as to what exactly practitioners are doing when they are practicing modern mindfulness-based interventions.” In short, “mindfulness” does not currently meet fundamental criteria required by science that can be clearly defined and operationalized [33]. Nevertheless, many studies have reported the effectiveness of mindfulness-based programs.

In 2014, the Mental Health Services of the Department of Veterans Services commissioned the Department of Psychiatry at the University of Rochester to conduct a multisite Meditation for PTSD Demonstration Project, headed by Dr. Kathi Heffner [17]. The current meta-analysis includes all these studies as well as all others that used a treatment identified as meditation found in a systematic search of the literature. The goal of this meta-analysis is to assess the effectiveness of various meditation techniques in the treatment of PTSD.

## 2. Methods and Materials

Our methodology was published in the Prospero International prospective register of systematic reviews, National Institute of Health and Care Research [34]. We conducted a multiple-treatments meta-analysis (Cochrane, 16.6.3) [35] that directly and indirectly compared the within-groups effects of four different categories of meditation using the Comprehensive Meta-Analysis (CMA) program Version 4, Meta-Regression 2 programs [36].

**Searches.** Following the Preferred Reporting Items for Systematic Reviews and Meta-Analyses guidelines for 2020 (Prisma, https://www.prisma.io/ (accessed on 3 December 2024)), we searched for papers in English, published and unpublished, on the use of meditation approaches for treating PTSD. Searches were made in major databases (MEDLINE, PubMed, PsycINFO, Web of Science, Library of Congress, and Google Scholar), as well as the bibliographies of systematic reviews and meta-analyses, original research papers, and research anthologies and databases of meditation research, from January 1970 through June 2024. The search included references in the most recent meta-analyses on meditation and PTSD, Hilton et al., 2017 [37], which reported 10 RCTs, and Gallegos et al., 2017 [16], which included 19 studies. We also asked colleagues in the field for relevant papers. A Prisma flow chart was constructed documenting the results of the literature search. Search terms included “Mindfulness-Based Stress Reduction”, “MBSR”, “Transcendental Meditation”, “TM”, “mindfulness”, and “meditation”.

**Participant populations.** We included all participants diagnosed with PTSD, all types of traumas, all age groups, all gender identifications, all ethnicities, and all countries. When data were missing on age, we used the mean age of their meditation category. This was for the purpose of multiple meta-regression analysis, in which one study with missing data would reduce the N for all comparisons. We included studies conducted on populations from VA facilities, refugee camps, prisons, medical clinics, school counseling services, and all other facilities providing treatment for PTSD.

**Types of studies included.** We included all longitudinal research designs that measured change over time that have sufficient data to enable calculation of effect sizes and variances at each measurement period. This included Randomized Controlled Trials (RCTs). We also included Controlled Trials (CTs) that were not randomized, and Single Group case series studies, which are single groups who underwent instruction in the intervention without a comparison or control group and were measured multiple times, at baseline and with at least one posttest.

**Studies not included.** When studies were reported in two or more publications, we used the latest publication. Heffner et al., 2014 Syracuse [17] was not used because it overlaps with Polusny et al., 2015 [38], which we did use. We did not include studies that did not report enough data that could be used to calculate or derive g (e.g., Bremner et al., 2011 [39], Ge et al., 2020 [40]). Malaktaris et al., 2022 [41] was not used because it was a hierarchal multiple regression analysis of subjects in Bormann et al., 2013 [42], and 2018 [43], not a study of change in PTSD due to a meditation treatment. Mistry et al., 2020 [44] (Virtual Reality Meditation, VRM) was not used because PTSD symptoms were used only to classify subjects. No longitudinal results of the effects of VRM on PTSD were reported. Nicotera et al., 2020 [45] was not used due to lack of data: “*our study did not have access to direct measures of PTSD*”. Sullivan et al., 2021 [46] was not used because of lack of data. The abstract reported that “*The sample size was insufficient for inferential analysis and the data for some participants was not collected at all-time points*”.

**Screening.** Two authors (DOJ, JT) working independently conducted the data searches, with updated searches made in 2020, 2021, 2022, 2023, and 2024. Two authors (DOJ, BR) independently coded studies for implementation statistics. Two authors (DOJ and VB) independently coded studies for research quality on the modified Clear Score scale. Data for calculating effect sizes and meta-regression were coded by DOJ and independently proofed by Rhoda Orme-Johnson, a professional proofreader. Any discrepancies between coders were discussed and resolved with reference to the original text in the papers.

**Treatments.** The four classifications of meditation treatment groups were defined as follows:Mindfulness-Based Stress Reduction (MBSR) is a manualized program with several components, involving both focused attention (Breath Awareness Meditation) and open monitoring practices, as well as yoga, retreats, and other components [30,47,48].The Transcendental Meditation technique (TM) falls in the category of automatic self-transcending [49]. It is a standardized meditation technique from the Vedic tradition of India brought to the West by Maharishi Mahesh Yogi in the 1950s [50,51,52]. TM is described as a simple, natural, effortless process of transcending from active mind to the silent mind. During TM, the mind is automatically drawn inward towards increasing levels of charm to arrive at a state of restful alertness, transcendental consciousness, a unique fourth state of consciousness that transcends the meditation process itself, hence the name “automatic self-transcending” [49].Mindfulness Based Other (MBO) techniques are meditations that identify themselves as mindfulness, or have been derived directly from MBSR, but are not specifically MBSR.Other Meditations (OMs) are techniques described as meditation but are neither TM nor mindfulness. Descriptions of the specific techniques can be found in the papers cited.

Mindfulness and TM were chosen for analysis because the most meditation research has been conducted on these two modalities [53]. OM was included to have all types of treatments called “meditation” in our meta-analysis. These meditations were all mental techniques. We did not include related physical techniques such as yoga asanas, although MBSR and some meditation programs include a yoga component.

**Comparator(s)/control.** We report the within-group effects of the four categories of meditation on treating PTSD. This provides a direct comparison of the different meditation categories without being obscured by the effects of controls. The “controls” were the three other categories of meditation to which each category was compared.

**Measures of PTSD.** Total PTSD symptoms were measured by the PCL symptoms checklist or measures equivalent to the PCL, such as the Clinician-Administered PTSD Scale (CAPS). The PCL is now a 20-item questionnaire (previously 17 items), corresponding to the DSM-5 symptom criteria for PTSD. The PCL is the most widely used measure of PTSD, and we use it for primary analysis because that allows inclusion of all studies in our meta-analysis. The Clinician-Administered PTSD Scale (CAPS), a semi-structured interview for determining PTSD diagnosis and symptom severity, was reported in only about half the studies. When only Cohen’s d was reported, we converted d into g, where the conversion of d to g is g = d times a correction factor J; J = 1 − 3/(4df − 1) [54].

**Special cases of computing effects**. When the PCL was not reported, we converted other measures on PTSD symptoms into g. Azad and Morteza, 2014 [55] reported d for the total quality of life score, which we converted to g. For Christopher et al., 2018 [56], which reports on police being treated with mindfulness meditation (REP-MM), we used the composite of measures of alcohol consumption, anxiety, depression, sleep difficulties, and suicidal ideation reported in the paper as the measure of PTSD. Omidi et al., 2013 [57] reported on veterans in Iran with PTSD, but PTSD was not measured pre-post. Six moods were measured (anger, dizziness, depression, fatigue, tension, and vitality), and we entered all of them in the analysis and used the g for the average change as the PTSD score. For Kim et al., 2013 [58] on nurses, we used the N, t, and *p* reported in the paper to calculate *g*. For Somohano et al., 2022 [59], PCL was not included because the authors reported it was not significant. We used the mean of the d’s given for the Emotional Tone measures, sadness, disgust, fear, and anger, then converted d to g. If the SDs were not reported, we estimated the *SE* (standard error) from other studies; that is, *SE* = 0.350.

**Other statistics reported.** In addition to pooled effects of g, we reported 95% confidence limits, using the random-effects model. Results are presented in forest plots with associated statistics on the summary effects across studies, weighted by inverse variance. The dispersion of effect sizes from study to study, the heterogeneity, measured by the prediction interval, is a major focus. Combining studies within subgroups, we did not assume a common among-study variance component across subgroups. We did not pool within-group estimates of tau-square. This is the option used by RevMan and the CMA software tool Version 4 used in this meta-analysis [35]. We used the fixed-effects model to combine subgroups to yield an overall effect.

We also extracted data and made tables on study characteristics, which included the location of the study, population type, trauma type, comparator or control, PTSD and other measures used, age, gender, sample size, longest follow-up, notes, and implementation outcomes, such as rate learned vs. offered and the rate of subjects completing the course.

**Risk of bias (quality) assessment**. Study quality was assessed by using a modified 15-item CLEAR score check list [60] assessing potential bias related to provider competence, non-parallel design, non-equivalent groups, and incomplete meditation learning and practice. The CLEAR score was the total number of items answered “Yes”. We also scored the studies using a 7-item CLEAR2 scale of research quality items that do not involve a control group. This was because two within-group analysis studies in this meta-analysis are the pretest to posttest effects on the treatment groups with no reference to control groups.

**Meta-Regression**. Regression analysis is a set of statistical methods used for estimating the relationships between an outcome (dependent) variable and a predictor (independent) variable. For example, in our meta-analysis, the dependent variable is the effectiveness of meditation techniques, as measured by the effect size of recovery from PTSD from pretest to posttest, expressed in Hedges’s g units. We wanted to know if moderator variables influenced the effectiveness of meditation on PTSD. Did the different types of meditation—MBO, MBSR, OM, and TM—have different effect sizes? What are the effects of age? Are younger subjects more responsive to meditation than older subjects? Does meditation have different effects on different trauma groups? For example, how much do combat soldiers, earthquake victims, war refugees, prison inmates, and intimate partner violence (IPV) survivors benefit from meditation therapy? What are the effect sizes of military personnel vs. civilians? Does research design (Randomized Controlled Trials (RCTs), Controlled Trials (CTs), Single Group) or research quality, study duration, baseline PTSD, or % males in the study modify effect sizes? To answer these questions, we used the Comprehensive Meta-Analysis V4, Meta-Regression 2 program [61].

**Outliers**. Two studies of the sixty-one studies in our meta-analysis had effects large enough to be considered as outliers. These were Rees et al., 2013 and 2014 [62,63], discussing the effects of TM on Congolese war refugees with PTSD in Uganda. The effects of g = −5.23 and −2.84 were three to five times larger than the mean effect of all other studies, large enough to distort statistical analyses. We omitted these studies from the main analysis but studied their effects. With these two outliers included, the overall effect of TM was g (SE) −1.31 (0.11), which was reduced to −1.13 (0.09) when outliers were excluded. The one-study-removed analysis found that removing Rees et al., 2013 [62] decreased the overall summary effect of TM from −1.31 to −1.19, with a *p*-value that was essentially zero.

The outliers increased indicators of heterogeneity. A key indicator of heterogeneity is the prediction interval, which shows the range of true effect sizes of 95% of all future studies on comparable populations. The inclusion of the outlier expanded the range of the prediction interval on the upper and lower ends. With the outliers included, the upper limit to lower limit was −2.32 to −0.30, a span of −2.02. Without the outliers, the span was −1.89 to −0.38, a span of 1.5, a reduction in the span by 25% by eliminating those two studies. I^2^, the measure of the between-group variance that reflects true differences between studies rather than sampling error, decreased from I^2^ = 91.28% with the outliers included to 86.17% with the outliers excluded. There was still a large amount of between-study variance needing explanation, which we addressed with meta-regression.

## 3. Results

**Literature Search.** Databases (MEDLINE, PubMed, PsycINFO, Web of Science, Library of Congress, and Google Scholar) were searched for keywords “Mindfulness-Based Stress Reduction”, “MBSR”, “Transcendental Meditation”, “TM”, “mindfulness”, and “meditation” from January 1970 to June 2024. These searches identified 9963 records on meditation. Of these, 8938 were excluded because they were not on PTSD or trauma or otherwise did not meet our study inclusion criteria. Titles and abstracts of 1025 records were obtained for screening, of which 907 were excluded, leaving 119 papers sought for retrieval and assessment for eligibility. The 119 records assessed for eligibility resulted in 62 records being excluded, because 14 were reviews, 8 were theoretical papers, and 14 records were not intervention studies on meditation. For example, they were feasibility studies on the use of meditation to treat PTSD or were cross-sectional studies, such as on the rate of PTSD in different populations or were correlational studies between the level of PTSD in people with different psychological or psychosocial characteristics, such as differences in qualities of mindfulness. Such studies, although of interest, did not address the issue of the efficacy of meditation as a treatment for PTSD and thus were excluded. The 26 others of the 118 were excluded because they were studies of the effects of other types of interventions in treating PTSD rather than meditation, such as trauma focus techniques, cognitive processing, hypnosis, hatha yoga, pharmaceuticals, or games. This left 57 studies from the searches of databases, which, with the addition of 4 TM studies identified via other sources, i.e., the Collected Papers on TM, resulted in a total of 61 studies included in our review.

Collected Papers on TM, which is on the righthand column in the flow diagram, refers to Volumes 1–8 of Scientific Research on the Transcendental Meditation and TM-Sidhi programs 23–28. These volumes, which span 1970 to 2013, contain 678 records. Of these, six records were on “PTSD” or “posttraumatic”, four met the inclusion criteria of a longitudinal intervention study on PTSD, which, added to the 57 studies identified via databases, resulted in 61 studies included in our meta-analysis.

These 61 studies were then logically divided into four categories of meditation, including the two categories that are uniformly defined, Transcendental Meditation (TM) and Mindfulness-Based Stress Reduction (MBSR). The first of the other two categories was Mindfulness-Based Other (MBO), which are mindfulness-based techniques but not MBSR. MBO techniques were often derived from MBSR but did not follow the MBSR protocol. MBO techniques often experimented with different lengths of training, mode of presentation (such as via telecommunications), or with additional techniques, such as Hatha Yoga. The fourth category of meditation was Other Meditations (OMs), which were neither TM nor MBSR nor MBO. Examples of OMs include compassion meditation, Kriya yoga, Adapted Mantra Meditation, and Mantram Repetition meditation.

Data were then extracted from these 61 papers on study characteristics, such as experimental design, type of treatment, number of subjects, type of trauma, location of treatment, etc. Implementation statistics were also coded, such as the number of people offered meditation, number and percent who learned, number and percent who were post-tested, etc. In addition, data from the papers needed for calculating effect sizes were entered into the meta-analysis and multivariate regression analysis program 35. Figure 1 displays the results of our literature search, which located 61 studies that used meditation to treat PTSD, 16 studies on MBO, 13 on MBSR, 14 on OM, and 18 on TM.

**Study Characteristics and Implementation Statistics.** Appendix A contains tables with the statistics of each of the studies in these four categories of meditation. There are four tables on study characteristics, one for each category, which alphabetically are Mindfulness-Based Other Meditation, Mindfulness-Based Stress Reduction, Other Meditation, and Transcendental Meditation, and four tables for meditation category with implementation statistics. The tables on study characteristics show the study identification and the location of where the study was conducted, the PTSD patient population, the research design (whether RCT, CT, or Single Group), the control group, the PTSD measures used, the baseline PTSD level on the PCL, the research quality on the CLEAR Score, mean age of the experimental group, the total sample size, the % males, and the study duration in weeks.

The tables on implementation statistics display the number of people the program was offered, the % who learned, number completed, percent completed, number of dropouts, percent dropouts, number and percent post-tested, and notes, if any.

Table 1 summarizes the study characteristics for each category of meditation: MBO, MBSR, OM, and TM.

The 61 studies had a total sample size of 3440 subjects: MBO = 636, MBSR = 813, OM = 743, and TM = 1248. The overall mean (SD) baseline level on the PCL was 52.2 (12.4), which was similar across all categories of meditation. A PCL score of 31–33 or higher indicates probable PTSD; 40 or higher allows for a provisional diagnosis of PTSD (if cluster criteria are met) [64,65]. A score of 50 is considered PTSD-positive in military populations [65]. The range of baseline PCL scores across all studies was from 29 to 75, ranging from borderline to severe PTSD. It was expected that these populations would have high levels of PTSD, because for most studies, clinical PTSD was a selection criterion for inclusion in the study.

The average person in these studies was middle aged (40–60), with an overall mean age of 46.8 (9.5) years. The mean age was very similar across meditation categories. A few studies were on young adults in their 20s, and we will present the effects of age on effect sizes in the section below on meta-regression.

The mean percent of males in the study populations was 65.1%. Across all categories on meditation, % males ranged from all women (0% males) to 100% males. No non-binary persons were identified as such in any of the studies. The maximum study durations ranged from 1 week to almost a year (48 weeks), with the average being 13.1 weeks. In the section on meta-regression, we present the results of % male and study duration on effect sizes.

We found that many different populations with PTSD are willing to try meditation. MBO has been used to treat male and female military personnel, most of whom had experienced combat, some whom experienced Military Sexual Trauma (MST). Other MBO studies were conducted with female survivors of trauma, police officers, nurses, students, women with PTSD plus substance abuse, and patients with ileostomy.

In treating PTSD, MBSR has been used primarily with military personnel, but also with breast cancer patients, child abuse survivors, women with a history of interpersonal trauma, and women experiencing PTSD from undisclosed causes. OM has also been used to treat PTSD in military personnel, including survivors of Military Sexual Trauma, and experiencing trauma from the sudden death of friends and comrades. OM has also been used to treat PTSD in traumatized youth in residential care. Most PTSD studies on TM have also been with military personnel, but it has also been used to treat PTSD in male and female prison inmates, women survivors of IPV, caregiver nurses, war refugees in Uganda and Ukraine, earthquake/tsunami victims in Japan, and South African college students.

**Implementation Statistics.** Table 2 summarizes the implementation statistics for the four meditation categories, which answers the following questions of the people who were offered meditation: what was the percentage of people who learned it (started the course), what percentage completed the course, and what percentage completed post-testing? Overall, 85% chose to learn, 81% completed the course, and 72% made it through to post-testing. (Appendix A).

**Research Quality.** There were 15 research quality items in the CLEAR score (Appendix A) The research quality items ask such questions as whether the study used validated measures of PTSD and were the same or equivalent measures used at pretest and posttest. It asked if the providers of the meditation technique were qualified and skilled, did the study have systematic follow-up and checking of correct practice of the meditation technique, and was participant adherence evaluated quantitively. Another modified scale, CLEAR2, was used to measure research quality on studies that did not involve a control group.

Appendix A shows scoring of the individual studies on research quality, as well as *t*-tests and a graph. The *t*-tests found there was no statistically significant difference between the studies in the meditation categories on research quality scores. The correlations of research quality of the four meditation categories across the 15 items were all positive and statistically significant, with r’s ranging from 0.60 to 0.93. This indicates that the strengths and weaknesses of research quality were similar for MBO, MBSR, OM, and TM. The lack of differences between types of meditation on research quality indicates that it is unlikely that research bias could cause differences in their effectiveness.

We note that in this meta-analysis of within-group effects, control groups were not an issue, because we used within-group standard mean differences pretest to posttest that do not involve control groups. The “control” groups were the other meditation categories. However, we did score all studies on CLEAR2, a scale of research quality items that do not involve a control group. The mean scores on this scale were MBO = 73.3%, MBSR = 68%, OM = 75.5, and TM = 77%; there were no statistically significant differences between groups.

**Effect sizes of the four categories of meditation.** Table 3, Table 4, Table 5 and Table 6 display the results for the 61 studies using the mixed-effects model, which is the appropriate model for studies known to differ from each other on attributes that could produce true differences between them (such as type of meditation, age, sample size, study duration, etc.) The statistics presented are the effect sizes in Hedges’s g units, standard errors, the upper and lower limits of the 95% confidence intervals, the Z-values, and *p*-values, as well as the summary statistics across studies at the bottom of each table. The overall effectiveness, in Hedges’s g (SE), was moderate for MBO g = −0.66 (0.10), MBSR g = −0.52 (0.05), and OM g = −0.63 (0.07) and large for TM g = −1.13 (0.09).

Table 7 summarizes the results for the mixed-effects model, giving the summary point values (mean g’s) and associated statistics. The mean effects of all categories of meditation are highly statistically significant (*p* < 0.00000001), indicating that all meditation techniques have real effects on reducing PTSD.

Table 8 shows the prediction interval and other heterogeneity statistics. Fixed effects are shown in the top half of the table, mixed effects below. In the upper right, “Other Heterogeneity Statistics” is essentially an analysis of variance table, which under the fixed-effects model assumes that differences between studies are only due to random sampling error. Q statistics indicate the amount of dispersion within the various groups under the fixed-effect model. The expected value of Q is the degrees of freedom of Q (df(Q)), which is the number of studies minus 1. It can be seen in Table 8 that for all meditation techniques, the Q’s are much larger than the df(Qs), indicating that the effect of each technique on decreasing PTSD is highly statistically significant, as shown by the *p*-values in the next column. The I^2^ (upper right in Table 8) indicates that the percentage of variance left unexplained is high for all categories of meditation, ranging from 56.82% to 86.23%, with 90% overall. We will explore sources of heterogeneity below using meta-regression analyses.

In the lower section of Table 8, under mixed-effects model, we see the Prediction Intervals, which are the indicators of heterogeneity. The Prediction Interval (PI) is the range of effect sizes that future studies on all comparable populations would likely fall into, if we assume that the true effects are normally distributed in g units. It is a common error in meta-analyses to believe that I^2^ is the measure of heterogeneity. It is not. I^2^ estimates the proportion of unexplained variance. PI estimates the spread of effects of studies, which is the heterogeneity. It is important to clarify what heterogeneity is in systematic reviews, because the PI may predict that some treatments are likely to have adverse effects, something that clinicians need to be aware of, which is not provided by the I^2^ statistic [107,108].

In the present meta-analysis, the prediction intervals indicate considerable heterogeneity for each of the meditation categories. The lower limits of the PIs show strong effects for each of the meditation categories, ranging from g = −0.84 for MBSR to −1.89 for TM. The upper limits of the PIs indicate the least effective results that future studies are likely to find. These range from −0.38 for TM, indicating that the least effective results of future studies on TM would be small, to a slightly positive value of 0.08 g for MBO. In this context, where we are looking for a reduction in PTSD, a positive PI indicates that the meditation technique may make PTSD worse. However, 0.08 only indicates slightly worse, essentially zero. For MBSR and OM, the upper limit of their PIs of −0.21 and −0.10, respectively, might indicate that some future studies are likely to find small effects that are not clinically useful.

The tau is the standard deviation of true effect sizes, and the tau-squared is the variance. We see that the variances for TM and MBO are approximately the same (0.34 and 0.33, respectively) and that the variance for MBSR and OM is smaller (0.13 and 0.23). This suggests that there is more variability in the outcomes for TM and MBO. Total between-groups Q, shown at the bottom of Table 8 (Q = 33.93, df(Q) = 3) indicates that there is a highly significant difference between the effectiveness of the four meditation categories on PTSD (*p* < 0.0000002).

### 3.1. Meta-Regression

The purpose of meta-regression is to assess what aspects of the research might help explain the unexplained differences between studies. Five principles behind constructing the best explanatory model are shown here.

Use the correct model, Random Effects, because these studies are known to differ on important dimensions, such as meditation treatment, age, type of trauma, research design, and other items that may influence effect sizes.Use the simplest model; too many variables will tailor the model to the specific data set and reduce the generalizability of the model.Maximize the R^2^ analog, which tells us what proportion of the true between-studies variance is explained by the model.Minimize the I^2^, the percent of true between-studies variance that remains unexplained after the model is applied.Maximize the statistical significance of the *p*-values of the components of the model.

Table 9 shows the regression statistics for the significant covariates, arranged from most predictive power (largest R^2^) to least. Table 10 shows the predictive power of combining the covariates into multivariate models by adding one covariate at a time.

In Table 9, it can see seen that Treatment Groups made the largest contribution, with an R^2^ = 0.41, followed by Age, Five Trauma groups, and Research Design. R^2^ became negligible for Study Duration and Baseline PTSD and zero for Military vs. Civilian, Percent Males, and Research Quality Score.

Table 10 shows the results of testing one covariate at a time, arranged from the largest R^2^ to the smallest. The statistics in each row give the result for that covariate controlling for the other covariates. For all covariates, the variance inflation factor (VIF) was less than 2.0, indicating low concern for multi-collinearity, which would negatively impact the precision of estimating model coefficients [54].

The top row in Table 10 shows the model with the intercept only, no covariates, which has no predictive power (R^2^ = 0). All models have the same intercept. The I^2^ indicates that 89.97% of the true variance remains unexplained with no covariates. Adding Treatment group (MBO, MBSR, OM, or TM) increased R^2^ from 0 to 0.41 and decreased I^2^ from 89.97% to 83.17%, and showed a change in I^2^ by −6.80%. Adding all the covariates to the model one at a time progressively increased R^2^ to 0.67 and reduced I^2^ to 72%, as seen at the bottom row of Table 10. All models were statistically significant at *p* < 0.0000000001.

Figure 2, Figure 3, Figure 4 and Figure 5 below show the scatter diagrams for the different covariates, controlling for all the other covariates. The circles represent the different 61 studies. The areas of the circles are inversely proportional to their variance. Larger circles represent larger and better controlled studies, which are given more weight in the meta-regression. The dark horizontal lines are the mean effects, and the lighter horizontal lines are the 95% confidence intervals.

Figure 2 shows a larger reduction in PTSD in the TM group than the other three meditation categories, which were not different from each other.

Figure 3 on ‘Age’ shows that most of the studies cluster in the middle-age range, from 40 to 60 years old, but there are a few studies on younger adults that pull the regression down on the left. Within the categories of meditation, age was significant for TM (*p* = 0.00001) and MBO (*p* = 0.03). Age was not significant for OM (*p* = 0.44) or MBSR (*p* = 0.50).

Figure 4 on “Research Design” shows that there is a greater mean reduction for CTs than for RCTs or Single Group studies, which did not significantly differ from each other.

Figure 5 on Trauma Groups suggests that prisoners showed greater reductions than other groups, but there were too few studies in the other categories to draw any conclusions. (See Appendix A).

### 3.2. Comparison of Military and Civilian Populations

Figure 6 shows the scatter diagrams for regression of Hedges’s g on Military vs. Civilian groups. Over all studies, meditation produced a greater mean reduction in PTSD for the studies on Civilians than for the studies on Military populations. Civilian groups also showed a wider dispersion of effects than Military groups, particularly on the low end, which indicates of greater reduction in PTSD.

### 3.3. Subgroup Analysis of Mantram Repetition (MR), MBSR, and TM in the U.S. Military

Mantram Repetition entails choosing a spiritual mantra from a list, such as the Sanskrit chant of “Om”, “Om Shanti”, or “So Ham”, Christian chants like “Hail Mary” or “Ave Maria”, or Buddhist chants like “Om Mani Padme Hum” [42,109,110]. We compared studies on Mantram Meditation (five studies) with the two most-studied meditation programs, MBSR (eight studies) and TM (eight studies). Mantram Repetition was included because the multisite demonstration project of meditation recommended it. “We recommend that meditation programs—and, in particular, the Mantram Repetition Program developed by Dr. Jill Bormann—are worthy of further in-depth assessment through both rigorously controlled randomized trials and comparative effectiveness studies [17], p. iii.”

The results (mixed effects, Hedges’s g (95% CI)) were MR −0.75 (−0.97, −0.54), MBSR −0.55 (−0.69, −0.40), and TM −1.06 (−1.17, −0.94). The between-group Q = 74.42, df (Q) = 2, and *p* = 0.0000000, indicating a significant difference between groups. The effect of TM was significantly larger than for MR (*p* = 0.015) or MBSR (0.00000008). MBSR and MR did not differ significantly at the 0.05 level (*p* = 0.12). Funnel plot analyses found no evidence of publication bias for all 21 studies together or for MR, MBSR, or TM analyzed separately. (See Appendix A.)

In comparing the effects of MBSR, TM, and Mantram Repetition in the U.S. military, N was 38 studies instead of 39, because we excluded the one non-U.S. study on the Israeli military (Fruchter, 2024 [98]). Subgroup analysis showed excluding Fruchter from the analysis only changed the effect of TM from 1.06 g to 0.99 g.

### 3.4. Comparison of MBO, MBSR, OM, and TM in Civilian and Military Studies

Figure 7 shows the overall results for this meta-analysis and the mean effect sizes in Hedges’s g units with 95% confidence intervals organized by the four types of meditations: MBO, MBSR, OM, and TM. Gold bars indicate the results for all 61 studies. Red bars indicate the results for the 38 studies on U.S. Military personnel. Green bars indicate the results for the 22 studies on Civilian populations.

The mean effect sizes are shown in white font at the bottom of each bar. All mean effects were statistically significant. The mean effects of MBO, MBSR, and OM were generally in the moderate effect range, and they did not differ significantly from each other at the *p* = 0.05 level (See Table 11 below). TM, on the other hand, had large mean effects of over 1.0 g for both Military and Civilian groups, and these were significantly larger than for each of the other meditation categories.

The details of these analyses are in Appendix A.

## 4. Discussion

The 61 studies in this meta-analysis (MA) are more than the previous MAs on meditation for treating PTSD, which ranged from 8 to 21 studies [16,37,111,112,113]. Previous MAs had fewer studies because they included only between-group effects from RCTs, whereas we included within-group effects from RCTs, CTs, and Single Group studies. Like previous MAs [16,111,112], we found no appreciable differences in the study characteristics of research conducted on different meditations in terms of the types of study populations included, outcome measures, control conditions, gender, or length of time between the intervention and assessment of PTSD. Previous MAs found no effect of age for mindfulness meditations [112]. Similarly, our meta-regression found only slight evidence for a larger effect with younger populations for TM and OM, but no age effects for MBSR or MBO. However, none of these studies were designed to study the effects of age, which we suggest will be important for future studies. Studies on the effects of age would need to be designed to have a systematic spread of age groups treated under the same or similar conditions, the same populations using the same meditation technique.

The pooled mean effect in Hedges’s g for all 61 studies was −0.67, a medium effect size. For the four categories, they were MBO = −0.66, MBSR = −0.52, OM = −0.63, and TM = −1.13. These values are based on within-group analyses and are somewhat larger than those reported in previous meta-analyses because previous ones were based on between-group analyses. All effect sizes were statistically significant at *p* < 0.0000001. The standardized mean difference (SMD) effect sizes for MBO, MBSR, and OM are medium, indicating that the changes in PTSD symptoms from pretest to posttest were of moderate magnitude.

The large treatment effect of TM, with a g of approximately 1.0, is also in accord with previous MAs. The Final Report to the Department of Veterans Affairs on the large multi-site comparison on meditation techniques for treating military personnel with PTSD stated “*Based on PTSD and other outcomes observed here, Transcendental Meditation™ (TM) may hold promise as an effective tool for managing PTSD symptoms in Veterans [17], p. 52*”. A recent scoping review said *“Meditation and mind–body–spirit interventions such as Transcendental Meditation have high-quality randomized controlled trials demonstrating equivalent effect sizes and efficacy to existing evidence-based treatments for PTSD. Transcendental Meditation specifically appears to have sufficient empirical support to be considered ‘evidence-based’” [114].* A 2022 MA of mantra-based meditations also noted larger effects of TM than other mantra-based techniques. It reported eight studies on PTSD, four with TM and four with Mantram Repetition (MR) [115]. It had two analyses of two TM and two MR studies each, which did not reach statistical significance. However, when we re-analyzed their data in one analysis of four TM and four MR studies, TM’s effect was significantly larger than MR’s; *p* = 0.01. (The results are available from the lead author, DOJ). This MA [115] concluded “*TM seems to produce greater effects than other mantra-based meditations (OMBMs). Some studies support the need to differentiate TM from OMBM, because TM allows the subject to reach a higher state of consciousness in which the mantra progressively becomes a secondary experience until its disappearance*” [116], p. 13. TM produces large effects of approximately 1.0, which suggests that the patient, their family, and attending medical staff are likely to have noticed and commented on the difference [54].

Our meta-regression found that Treatment Group, Age, Trauma Groups, Research Design And Military vs. Civilian study populations were significant covariates in the regression on effect size (Hedges’s g). Yet even the best multivariate meta-regression model of these variables only reduced the unexplained variance (I^2^) to 72%. There is still much to be learned about what factors can enhance the effectiveness of meditation treatments, and we hope that our study will inspire new avenues of research.

### 4.1. Differences Among Meditation Techniques

There are many differences between TM and mindfulness-based and other meditation techniques or procedure that could account for TM’s greater effectiveness in treating posttraumatic stress. Procedurally, there are three different types of meditation: focused attention, open monitoring, and automatic self-transcending [49]. Meditation procedures described as “Mindfulness” are either “open monitoring” (OM), the non-reactive monitoring of the moment-to-moment content of experience, or focused attention (FA), which entails voluntary and sustained attention on a chosen prescribed sensory object, mental thought, or physiological process [49]. TM practice is characterized as an automatic self-transcending technique (AST), in which the mantra is repeated in an effortless way as it spontaneously becomes more refined until it transcends to silent mind: no mantra, no thoughts, just pure consciousness. In TM, the mantras assigned by the TM instructors originate from the ancient Vedic tradition of India. They are understood from tradition to have positive effects on all levels of the mind. The beneficial effects for such subtle experience is supported by findings in over 700 scientific studies [23,24,25,26,27,28]. Instruction in the technique, which entails selection of the appropriate mantra for the pupil and instruction in how to use it properly, is carried out according to tradition [50,51,52,117]. Maharishi Mahesh Yogi, who introduced the technique to the population at large, describes it as follows: “*The Transcendental Meditation technique is an effortless procedure for allowing the excitations of the mind gradually to settle down until the least excited state of mind is reached. This is a state of inner wakefulness with no object or thought or perception, just pure consciousness aware of its own unbounded nature. It is wholeness, aware of itself, devoid of differences [118]*”. In comparison, Bhante Henepola Gunaratana, a Sri Lankan Theravada Buddhist monk, who has published 14 books on meditation and mindfulness, describes mindfulness as follows: “*Meditation is not easy. It takes time and it takes energy. It also takes grit, determination, and discipline. It requires a host of personal qualities that we normally regard as unpleasant and like to avoid whenever possible. We can sum up all of these qualities in the American word gumption. Meditation takes gumption*” [119].

### 4.2. Proposed Mechanism for Reduction of PTSD Symptomatology Through TM Practice

The first physiological research on TM in the early 1970s attests to its effortlessness. It found that TM produces effects that are opposite to the fight-or-flight stress response [19,20,21,22]. This physiological state came to be referred to as a restfully alert, hypometabolic, physiologic state [20], or “restful alertness”. Whereas PTSD can be associated with increased heart rate, rapid breathing, muscle tension, pupil dilation, or increased blood pressure, effects of TM tend to be the opposite. A meta-analysis of 32 studies found that, during TM, respiratory rate, skin conductance, and plasma lactate level significantly decrease compared to controls resting with eyes closed [120]. This meta-analysis [120] also found that baseline levels of heart rate, spontaneous electrodermal responses, plasma lactate, and respiratory rate were lower prior to TM practice sessions than they were in controls prior to eyes-closed rest, suggesting long-term effects likely to reduce hyperarousal in PTSD patients. Moreover, TM subjects have been found to exhibit significantly faster recovery from stress than controls in electrodermal [22], cardiovascular [121], and cortisol responses, consistent with the finding of rapid recovery from PTSD. An magnetic resonance imaging (MRI) study found that activity in the brainstem, which regulates breath rate, heart rate, and cortisol responses, decreased while frontal-lobe executive blood flow increased, suggesting increased control at a deeper level of awareness during TM [122]. Also, whereas PTSD is treated with drugs that increase serotonin to improve mood [8], TM is associated with increased excretion of the serotonin metabolite 5-hydroxyindoleacetic acid (5-HIAA) and decreased excretion of adrenaline and noradrenaline metabolites [123,124,125]. These differences in neurotransmitter metabolism suggest a shift to a lower state of arousal, consistent with a normalizing effect. This may represent a shift to the parasympathetic side of the balance between sympathetic and parasympathetic components of the autonomic nervous system. Such a shift might underlie reduction of the hyperarousal associated with PTSD that occurs after beginning meditation practice.

Recent studies of effects of meditation on gene expression in peripheral blood mononuclear cells (PBMC) may support this earlier work on neurotransmitter metabolism. Long-term practitioners of TM show reduction of the Conserved Transcriptional Response to Adversity (CTRA) in PBMC [126]. The CTRA is mediated by a β_2_-adrenergic-receptor (i.e., sympathetic nervous system) mediated transcriptional response to chronic adverse conditions or stressors and is associated with increased inflammation and reduced resistance to infectious diseases [127]. The CTRA also was found to be reduced in a short-term randomized controlled trial involving another technique of meditation somewhat similar to TM [128]. To date, there have been no comparative effectiveness studies of effects of different meditation techniques on the CTRA gene expression profile.

Meta-analyses indicate that TM is superior to other methods of stress-reduction in decreasing blood pressure [129] and anxiety [130,131]. Similarly, a major review found that cardiovascular risk factors were lower in TM adolescents and adults than controls receiving various other treatments [132]. Repeated systemic reduction of the fight-or-flight response could be expected to be beneficial to persons suffering from hyperarousal. In comparison with TM, MBSR [118] includes both OM and FA tasks of breath awareness meditation (BAM), which requires the focusing of attention, which may keep the mind from transcending. Hence MBSR would require reduction of hyperarousal via mechanisms other than those resulting from transcending.

The initial distinction between the three categories of meditation is based on their different effects on EEG frequency [49]. AST is associated with increased alpha1 frequency (8–10 Hz), which coordinates the activities of distal cortical areas that are needed for memory, comprehension, creativity, and fine motor coordination [133,134]. OM increases theta frequency (4–6 Hz) EEG, which is associated with inhibition of sensory input and is needed to reduce distraction during internal mental processing like mental arithmetic or open monitoring meditation. FA is associated with increased beta/gamma frequency (20–50 Hz), which coordinates the activities of adjacent cortical areas needed for focused attention [49].

AST, OM, and FA also have different effects on EEG coherence, which is the stability of phase angle between EEG signals from different regions of the cortex. During OM, theta and alpha2 (10–12 Hz) coherence increases, and this is associated with idling of the visual cortex. During TM, alpha1 (8–10 Hz) has consistently been found to become coherent [135] among all cortical leads. TM also increases EEG synchrony, which is another way to measure the correlation of brain waves [136].

Increased alpha1 coherence observed during TM is correlated with creativity [137], concept learning [138], decreased anxiety [139], moral reasoning [139,140], neurological efficiency [138], a consciousness orientation [139], and higher states of consciousness [137]. These cognitive improvements could be expected to help the PTSD sufferer regain the ability to concentrate and improve their mood and sense of self-worth.

Different meditation techniques also have different effects on the Default Mode Network (DMN) [141], a system of the medial prefrontal and posterior cingulate cortex that becomes active when the mind is self-referral, concerned with issues of “me”, “my”, and “myself”, such as when describing “who I am”, or “what I think about something or someone” [142]. The DMN has been described as the “genius lounge”, in which the major systems of the brain responsible for memory, emotions, and decision-making sit together in a lounge to discuss and try to find solutions to the person’s current concerns [143]. The DMN works in the background on the preconscious level of the mind, and as solutions are found, they break forth into conscious awareness, sometime dramatically as “Aha” moments of deep insight. If DMN cannot find a solution to a deeply troubling personal problem, as in PTSD, the mind may become stuck ruminating in DMN mode, which explains why extended DMN activity is found associated with depression [144]. Persons suffering from PTSD commonly confront existential questions as a result of their traumatic experience, such as why did this happen? How could God allow such a thing? How can I give dignity to what happened? Do I deserve to be here? How can I resolve the guilt and shame I feel from what I saw, or did, or failed to do, etc.? Such irreconcilable conflicts often cannot be resolved on the level of talk therapy or reframing or figuring things out; lack of resolution can result in depression and suicide. Anecdotal reports from veterans and first responders indicate that even a brief transcendental experience of an inner field of awareness that is beyond conflict can ignite an optimism that life beyond PTSD is a possibility [145].

During TM, when the mind is effortlessly drawn inward due to the increasing charm at subtler levels, the default mode network (DMN) is initiated, as indicated by increased power and coherence of its characteristic frequency, alpha1, and by neuroimaging [141]. In contrast, FA and OM decrease DMN activity because they draw attention outward to accomplish tasks [141].

Mindfulness has been found to increase cortical thickening of areas of the brain that are exercised by mindfulness practices [146], whereas TM does not increase cortical thickening [122,147]. The brain, like muscles, increases in thickness with exercise, and the fact that TM does not increase cortical thickness may be an indicator of its effortlessness. Rather, TM increases cortical integration, as reviewed above, as well as integration of deep brain structures correlated with decreased stress.

A three-month-long study using resting-state functional MRI found that TM decreased perceived anxiety and stress, which correlated negatively with changes in functional connectivity among posterior cingulate cortex (PCC), precuneus, and the left superior parietal lobule. The PCC is a key node in the DMN, associated with emotion, cognition, awareness, and arousal. The precuneus is involved in self-processing operations, such as taking a first-person perspective, as well as in attention, memory, motor and mental imagery. The left superior parietal lobule is involved in spatial localization in addition to visual perception, reasoning, working memory, and attention. TM was also associated with increased connectivity between the PCC and the right insula, which plays a key role in interoceptive awareness (feeling internal bodily states like pain, hunger, and heart rate), emotional processing, empathy, and the perception of salient stimuli, particularly related to negative emotions, with a strong focus on the body’s internal state and self-referential processing. It is also involved in attention control and decision-making by integrating sensory information with emotional context. These results may help explain how the experience of restful alertness through TM integrates cortical and subcortical systems to heal the multidimensional breakdowns associated with PTSD.

### 4.3. Limitations

There were few direct comparisons of the different meditation techniques on the study level, which could have provided better controls for the duration and frequency of meditation sessions, the meditation teacher or instructor, participant selection criteria (including baseline psychological state and type of trauma), the environment where meditation takes place, the level of participant engagement, and the control group design (ensuring it provides similar attention and social interaction as the meditation group). Also, the sample sizes were relatively small, making it difficult to determine if the results generalize to the wider populations.

Everyone has a worldview born of their life experiences, which potentially could be a source of bias in scientific research. We used funnel plot analyses, which found no evidence of publication bias for any category of meditation. We included all the studies we could locate, which were more studies than previous meta-analyses. This is because our selection criteria were more inclusive, and our study was up to date. It included many recent studies that had not been reviewed before. To be transparent, we have presented the data for each individual study in Appendix A. The summary effects for MBSR, MBO, OM, and TM were similar in magnitude to those reported in previous systematic reviews by researchers independent of MIU or TM, giving confidence that our results were not biased towards overreporting TM nor underreporting other techniques. Furthermore, the effects for studies conducted at institutions independent of MIU of TM practice were of similar magnitude to the effects of studies that are associated with TM and MIU, as have been found in a previous meta-analysis on the psychological effects of meditation [148].

### 4.4. Future Research

In coding these studies, we found that implementation statistics on dropouts, completion rates, and regularity of meditation practices were not reliably reported across studies. This makes it difficult to quantitatively study the effects of these variables on treatment outcomes. We suggest that future research make sure to include these measures, which may be useful in finding ways to improve compliance and effectiveness of meditation in the treatment of PTSD.

We suggest also adding measures of transcending to all studies on meditation. We have made a case that transcending is the key to the effectiveness of TM. We suggest that bio markers of transcending, such as slowing of breath, reduced spontaneous skin resistance responses, and/or alpha1 (7–10 Hz) EEG power and coherence, could be measured along with any meditation practice to see if they produce a degree of transcending and whether this is predictive of efficacy in reducing symptoms of trauma. Are there other meditation techniques, or different forms of known ones, that enable transcending?

There has been much discussion and research on mindfulness as the key ingredient of successful meditation. Now we are suggesting a different paradigm, with the study of transcending as potentially the key ingredient. In this meta-analysis, we have argued that transcending is the reason that TM is successful, and not transcending is the reason that other meditation techniques are of limited efficacy. Perhaps a systematic search for transcending will lead to discovery of other meditation techniques that result in it.

The authors are currently preparing a second study which examines the effects of TM on PTSD in both military and civilian populations compared to controls. In addition, the main recommendation from this MA is that multisite Phase 3 clinical trials are needed to test TM’s efficacy compared with standard treatments, such as Nidich et al., 2018 [104], to help with the management of this public health problem of global proportion. Stakeholders among those serving veterans would benefit from studies that could help elucidate the roles for other meditation techniques as well.

## 5. Conclusions

This systematic review and meta-analysis of all meditation techniques used for treating PTSD found that all groups of techniques show benefit. The Transcendental Meditation technique has clinically meaningful effects on reducing PTSD symptoms that are statistically larger than for other meditation techniques. For this reason, and because TM is offered on a voluntary basis, as is the case with any evidence-based modality, it should be made widely available to military and civilian populations who desire to learn it. This research suggests that the wide implementation of this very effective treatment for PTSD would have significant cost savings and clinical implications, including reduced symptom severity, improved quality of life, increased functional ability, decreased reliance on medication, better social relationships, and potentially a greater capacity for resilience in individuals experiencing post-traumatic stress disorder, essentially allowing people with PTSD to manage their symptoms and participate more fully in daily life.

## Figures and Tables

**Figure 1 medicina-60-02050-f001:**
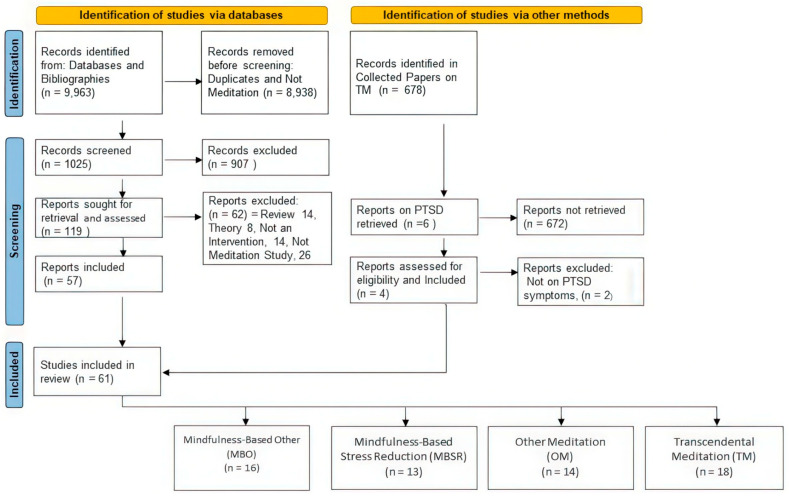
Flow diagram of search of the literature.

**Figure 2 medicina-60-02050-f002:**
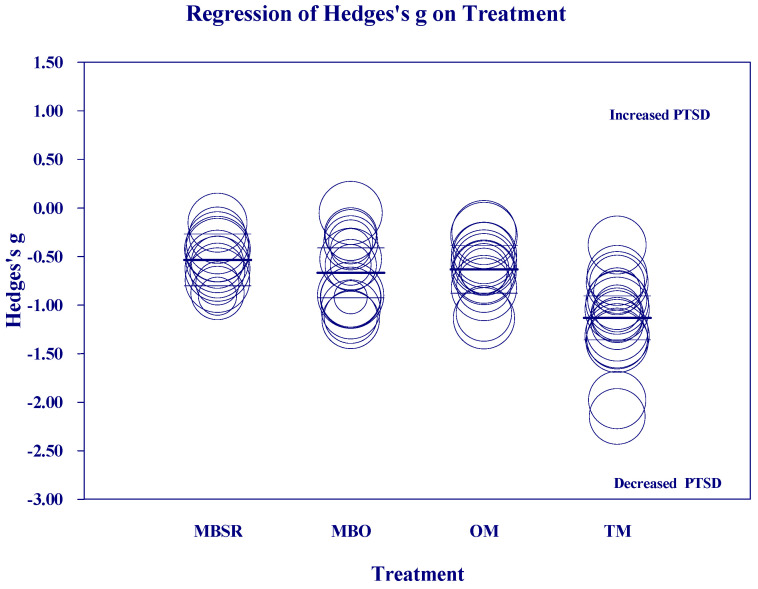
Regression of Hedges’s g on meditation Treatment group.

**Figure 3 medicina-60-02050-f003:**
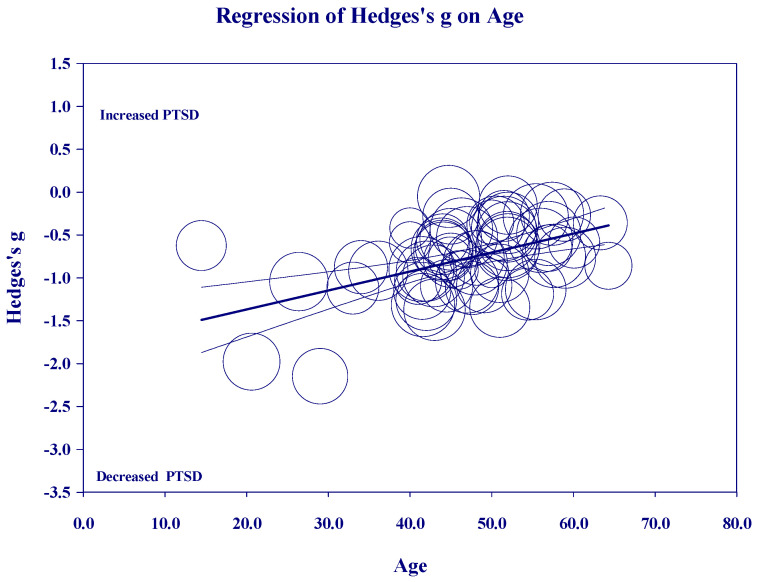
Regression of Hedges’s g on Age.

**Figure 4 medicina-60-02050-f004:**
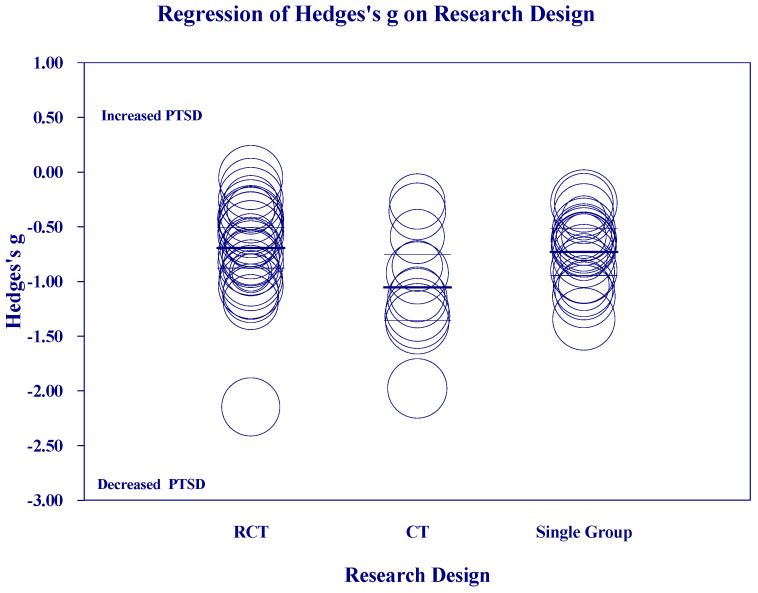
Regression of Hedges’s g on Research Design.

**Figure 5 medicina-60-02050-f005:**
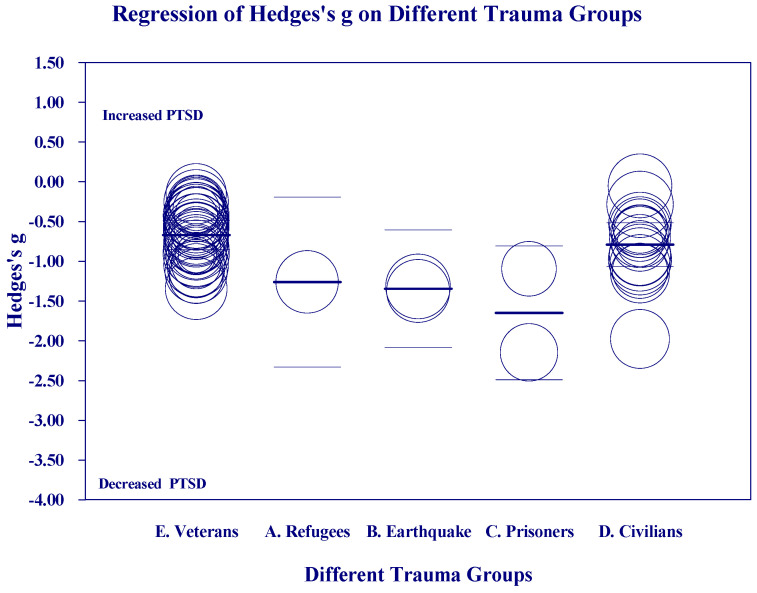
Regression of Hedges’s g on Trauma Groups.

**Figure 6 medicina-60-02050-f006:**
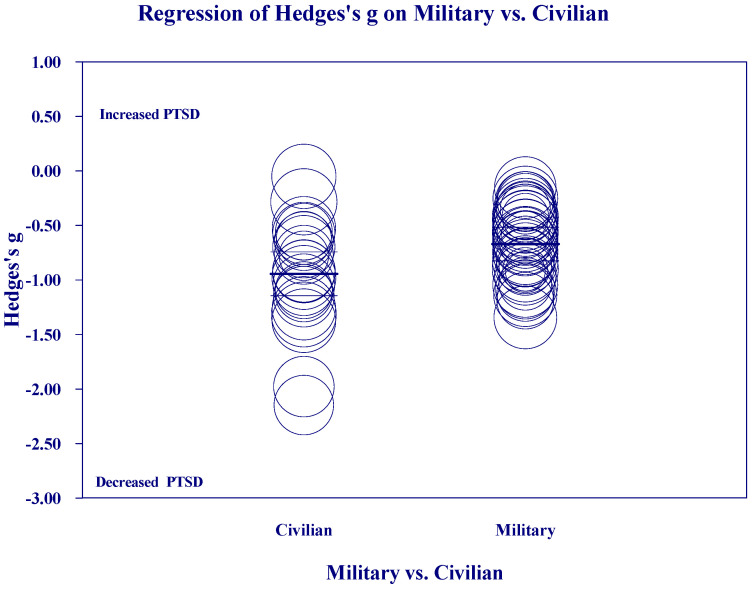
Regression of Hedges’s g on Military vs. Civilian studies.

**Figure 7 medicina-60-02050-f007:**
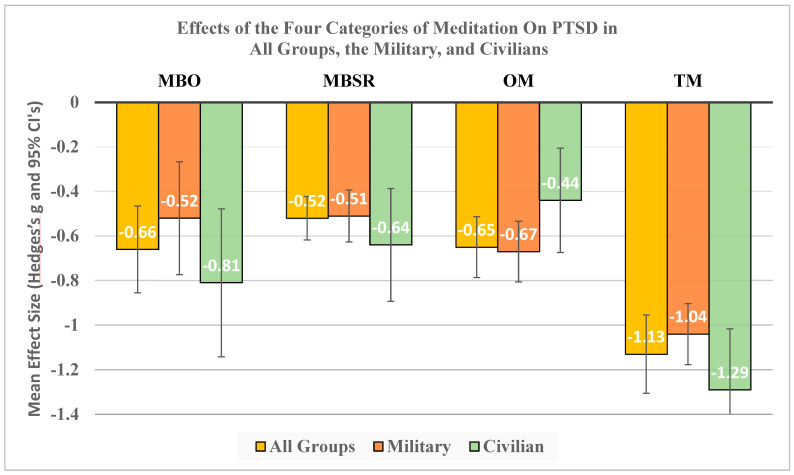
Hedges’s g with 95% CI for the four meditation categories in treating PTSD for All Groups, Military, and Civilian groups.

**Table 1 medicina-60-02050-t001:** Summary of study characteristics in each category of meditation.

Meditation Technique	Statistic	Baseline PTSD	Age (Years)	Sample Size	% Males	Max Study Duration (wk)
MBO, N = 16	Mean	53.3	46.9	45.4	60.1	10.2
	SD	13.5	9.8	40.2	42.8	4.9
	Range	30–75	26–64	10–176	0–100	8–24
MBSR, N = 13	Mean	52.5	48.1	57.4	66.2	15.1
	SD	10.6	5.6	52.7	40.9	6.5
	Range	35–63	34–58	9–116	0–100	8–24
OM, N = 14	Mean	52.2	49.4	53.1	76.5	14.9
	SD	11.0	11.5	53.1	28.6	14.8
	Range	35–66	34–58	15–184	16–100	4–48
TM, N = 18	Mean	50.6	42.9	65.7	57.6	12.7
	SD	14.6	11.0	68.7	40.0	5.4
	Range	29–66	21–62	5–239	0–100	1–24
Overall, N = 61	Mean	52.2	46.8	55.4	65.1	13.2
	SD	12.4	9.5	53.7	38.1	7.9
	Range	29–75	21–64	5–249	0–100	1–48

MBO = Mindfulness-Based Other Meditation, MBSR = Mindfulness-Based Stress Reduction, OM = Other Meditation, and TM = Transcendental Meditation.

**Table 2 medicina-60-02050-t002:** Summary of implementation statistics for the studies in each meditation category.

Meditation Technique	% Learned	% Completed	% Post-Tested
**MBO**	74%	64%	63%
**MBSR**	81%	71%	70%
**OM**	88%	85%	57%
**TM**	92%	89%	88%
**Overall**	85%	81%	72%

MBO = Mindfulness-Based Other Meditation, MBSR = Mindfulness-Based Stress Reduction, OM = Other Meditation, and TM = Transcendental Meditation.

**Table 3 medicina-60-02050-t003:** Results for Mindfulness-Based Other (MBO).

Mindfulness-Based Other (MBO)	Hedges’s g	Standard Error	Variance	Lower Limit	Upper Limit	Z-Value	*p*-Value
**Christopher, 2018 [56] MBRT**	−0.05	0.10	0.01	−0.25	0.15	−0.51	0.6130
**Grupe, 2021 [66] MBRT**	−0.53	0.12	0.01	−0.76	−0.29	−4.46	0.0000
**Heffner, 2014 [17] Houston IRV**	−0.28	0.23	0.05	−0.73	0.17	−1.21	0.2246
**Heffner, 2014 [17] Richmond MMIP**	−0.27	0.21	0.05	−0.69	0.15	−1.26	0.2069
**Heffner, 2014 [17] Richmond MMTH**	−0.36	0.20	0.04	−0.75	0.03	−1.82	0.0688
**Heffner, 2016 [67] New York Brief Mindfulness**	−0.35	0.31	0.10	−0.96	0.26	−1.13	0.2589
**Heffner, 2016 [67] South Carolina MBPT**	−0.92	0.47	0.22	−1.84	0.00	−1.96	0.0498
**Kelly, 2016 [68] TIMBSR**	−1.15	0.16	0.03	−1.47	−0.83	−7.07	0.0000
**Kim, 2013 [58] MBSB**	−1.12	0.21	0.04	−1.53	−0.71	−5.40	0.0000
**King, 2013 [69] MBCT**	−0.59	0.23	0.05	−1.04	−0.14	−2.59	0.0097
**Kirk, 2022 [70] MM + Yoga**	−1.04	0.15	0.02	−1.33	−0.76	−7.13	0.0000
**Norman, 2020 [71] MBCBT Vets**	−0.59	0.35	0.12	−1.27	0.10	−1.68	0.0939
**Norman, 2020 [71] MBCBT WW**	−0.42	0.36	0.13	−1.13	0.29	−1.17	0.2433
**Shin, 2021 [72] REP-MM**	−0.92	0.13	0.02	−1.17	−0.67	−7.25	0.0000
**Somohano, 2022 [59] MBRP**	−0.92	0.14	0.02	−1.19	−0.65	−6.69	0.0000
**Zalta, 2018 [73] MBSR + Yoga**	−0.90	0.05	0.00	−0.99	−0.80	−18.6	0.0000
**Mean Effect Size MBO**	−0.66	0.10	0.01	−0.86	−0.47	−6.73	0.0000
**Prediction Interval MBO**	−0.66			−1.41	0.08		

**Table 4 medicina-60-02050-t004:** Mindfulness-Based Stress Reduction (MBSR).

	Hedges’s g	Standard Error	Variance	Lower Limit	Upper Limit	Z-Value	*p*-Value
**Azad, 2014 [55] MBSR**	−0.91	0.39	0.15	−1.67	−0.15	−2.35	0.0186
**Bremner, 2017 [74] MBSR**	−0.88	0.21	0.04	−1.28	−0.47	−4.24	0.0000
**Cole, 2015 [75] MBSR**	−0.28	0.17	0.03	−0.62	0.06	−1.62	0.1060
**Davis, 2018 [76] MBSR**	−0.43	0.06	0.00	−0.54	−0.31	−7.11	0.0000
**Gallegos, 2015 [77] MBSR**	−0.56	0.22	0.05	−0.99	−0.13	−2.57	0.0102
**Goldsmith, 2014 [78] MBSR**	−0.65	0.19	0.04	−1.02	−0.27	−3.38	0.0007
**Kearney, 2012 [79] MBSR**	−0.63	0.08	0.01	−0.77	−0.48	−8.30	0.0000
**Kearney, 2013 [80] MBSR**	−0.36	0.11	0.01	−0.58	−0.14	−3.18	0.0015
**Kimbrough, 2010 [81] MBSR**	−0.78	0.32	0.10	−1.41	−0.15	−2.43	0.0150
**Niles, 2011 [82] MBSR**	−0.15	0.15	0.02	−0.44	0.14	−1.03	0.3024
**Omidi, 2013 [57] MBSR**	−0.55	0.11	0.01	−0.76	−0.34	−5.10	0.0000
**Polusny, 2015 [38] MBSR**	−0.74	0.08	0.01	−0.91	−0.58	−8.96	0.0000
**Possemato, 2016 [83] MBSR**	−0.43	0.10	0.01	−0.62	−0.24	−4.49	0.0000
**Mean Effect Size MBSR**	−0.52	0.05	0.00	−0.63	−0.42	−9.68	0.0000
**Prediction Interval MBSR**	−0.52			−0.84	−0.21		

**Table 5 medicina-60-02050-t005:** Other Meditation (OM).

	Hedges’s g	Standard Error	Variance	Lower Limit	Upper Limit	Z-Value	*p*-Value
**Bayley, 2022 [84] SKY**	−0.25	0.09	0.01	−0.43	−0.08	−2.88	0.004
**Bormann, 2008 [85] MR**	−0.65	0.28	0.08	−1.20	−0.10	−2.33	0.0196
**Bormann, 2013 [42] MR**	−0.48	0.07	0.00	−0.62	−0.35	−6.88	0.0000
**Bormann, 2018 [43] MR**	−0.83	0.07	0.00	−0.96	−0.69	−11.9	0.0000
**Church, 2020 [86] Eco Meditation**	−0.28	0.05	0.00	−0.38	−0.19	−5.79	0.0000
**Heffner, 2014 [17] Loma Linda MR**	−1.14	0.12	0.02	−1.38	−0.90	−9.26	0.0000
**Heffner, 2014 [17] Saginaw AMM**	−0.47	0.12	0.02	−0.71	−0.22	−3.79	0.0001
**Heffner, 2014 [17] San Diego MR**	−0.66	0.08	0.01	−0.82	−0.51	−8.39	0.0000
**Kearney, 2021 [87] LKM**	−0.56	0.06	0.00	−0.69	−0.44	−8.85	0.0000
**Lang, 2019 [88] LKM**	−1.09	0.18	0.03	−1.45	−0.73	−5.92	0.0000
**Lang, 2020 [89] LKM**	−0.66	0.11	0.01	−0.88	−0.44	−5.93	0.0000
**Schuurmans, 2020 [90] VRM**	−0.62	0.24	0.06	−1.10	−0.15	−2.60	0.0095
**Seppala, 2014 [91] Kriya Yoga**	−0.80	0.19	0.04	−1.18	−0.42	−4.10	0.0000
**Vasudev, 2020 [92] SKY**	−0.57	0.14	0.02	−0.84	−0.30	−4.14	0.0000
**Mean Effect Size OM**	−0.63	0.07	0.01	−0.77	−0.49	−8.83	0.0000
**Prediction Interval OM**	−0.63			−1.16	−0.10		

**Table 6 medicina-60-02050-t006:** Transcendental Meditation technique.

	Hedges’s g	Standard Error	Variance	Lower Limit	Upper Limit	Z-Value	*p*-Value
**Bandy, 2019 [93] TM**	−1.98	0.17	0.03	−2.30	−1.65	−11.9	0.0000
**Bellehsen, 2021 [94] TM**	−0.95	0.15	0.02	−1.24	−0.66	−6.40	0.0000
**Bonamer, 2019 [95] TM**	−0.70	0.14	0.02	−0.96	−0.43	−5.10	0.0000
**Bonamer, 2023 [96] TM**	−0.98	0.11	0.01	−1.19	−0.77	−9.11	0.0000
**Brooks, 1985 [18] TM**	−1.12	0.22	0.05	−1.55	−0.69	−5.09	0.0000
**Didukh, 2023 [97] TM**	−1.26	0.12	0.01	−1.49	−1.03	−10.6	0.0000
**Fruchter, 2023 [98] TM**	−0.38	0.16	0.03	−0.69	−0.07	−2.42	0.0157
**Heffner, 2014 [17] Minneapolis TM**	−0.86	0.27	0.08	−1.40	−0.32	−3.14	0.0017
**Heffner, 2014 [17] Saginaw TM**	−1.19	0.22	0.05	−1.62	−0.76	−5.44	0.0000
**Herron, 2017 [99] TM**	−1.34	0.13	0.02	−1.60	−1.09	−10.4	0.0000
**Kang, 2018 [100] TM**	−0.77	0.12	0.01	−1.00	−0.54	−6.56	0.0000
**Leach, 2023 [101] TM**	−0.77	0.19	0.04	−1.14	−0.40	−4.04	0.0001
**Nidich, 2016 [102] TM**	−2.15	0.18	0.03	−2.50	−1.79	−11.9	0.0000
**Nidich, 2017 [103] TM**	−1.09	0.21	0.05	−1.51	−0.67	−5.11	0.0000
**Nidich, 2018 [104] TM**	−1.04	0.08	0.01	−1.21	−0.88	−12.33	0.0000
**Rosenthal, 2011 [105] TM**	−1.04	0.27	0.08	−1.57	−0.50	−3.77	0.0002
**Yoshimura, 2015 [106] Ishinomaki TM**	−1.37	0.11	0.01	−1.59	−1.16	−12.3	0.0000
**Yoshimura, 2015 [106] Sendai TM**	−1.32	0.09	0.01	−1.48	−1.15	−15.4	0.0000
**Mean Effect Size TM**	−1.13	0.09	0.01	−1.31	−0.95	−12.5	0.0000
**Prediction Interval TM**	−1.13			−1.89	−0.38		
**All 61 Techniques: Overall Mean Effect Size**	−0.67	0.04	0.00	−0.74	−0.60	−18.4	0.0000
**All 61 Techniques: Overall Prediction Interval**	−0.67			−1.36	0.03		

MBO = Mindfulness-Based Other Meditation, MBSR = Mindfulness-Based Stress Reduction, OM = Other Meditation, and TM = Transcendental Meditation (See Appendix A).

**Table 7 medicina-60-02050-t007:** Summary statistics of the effect sizes of the different meditation categories and overall (mixed-effects model).

Mixed-Effects Model	Number of Studies	Mean Effect (g)	Standard Error	Variance	95% CI Lower Limit	95% CI Upper Limit	Z-Value	*p*-Value (2-Tail)
MBO	16	−0.66	0.10	0.01	−0.86	−0.47	−6.73	0000000
MBSR	13	−0.52	0.05	0.003	−0.63	−0.42	−9.68	0000000
OM	14	−0.63	0.07	0.01	−0.77	−0.49	−8.83	0000000
TM	18	−1.13	0.09	0.01	−1.31	−0.95	−12.55	0000000
Overall	61	−0.67	0.04	0.0013	−0.74	−0.60	−18.45	0000000

**Table 8 medicina-60-02050-t008:** Prediction Interval and Other heterogeneity statistics.

Group	Prediction Interval		Between-Study		Other Heterogeneity Statistics			
	Lower Limit	Upper Limit	Tau	TauSq	Q-Value	df (Q)	*p*-Value	I-Squared
Fixed effect								
MBO					94.35	15	0.000000	84.10
MBSR					27.79	12	0.0059	56.82
OM					94.37	13	0.000000	86.23
TM					122.92	17	0.000000	86.17
Total within					339.44	57	0.000000	
Total between					260.33	3	0.000000	
Overall					599.77	60	0.000000	90.00
Mixed effects								
MBO	−1.41	0.08	0.33	0.11				
MBSR	−0.84	−0.21	0.13	0.02				
OM	−1.16	−0.10	0.23	0.05				
TM	−1.89	−0.38	0.34	0.12				
Total between					33.93	3	0.0000002	
Overall	−1.36	0.03	0.35	0.12				

**Table 9 medicina-60-02050-t009:** Statistics on the effects of covariates: *p*-value of the covariate, I^2^ and R^2^.

Covariate	*p*-Value	I^2^	R^2^
Treatment Group	0.00000075	83.17%	0.41
Age	0.0000015	85.12%	0.35
Five Trauma Groups	0.00007	86.58%	0.26
Research Design (RCT, CT, SG)	0.013	87.5%	0.19
Study Duration	0.13	89.21%	0.07
Baseline PTSD	0.098	88.00%	0.01
Military vs. Civilian	0.008	89.93%	0.00
% Males	0.07	89.89%	0.00
Research Quality (Clear Score)	0.66	90.03%	0.00

**Table 10 medicina-60-02050-t010:** Results of multivariate meta-regression.

Model	R^2^	Change in R^2^	I^2^	Change in I^2^
Intercept Only	0	0	89.97%	0
Treatment	0.41	0.41	83.17%	−6.80%
Treatment + Age	0.58	0.17	77.89%	−5.28%
Treatment + Age + Trauma Groups	0.64	0.06	74.31%	−3.58%
Treatment + Age + Trauma Groups + Research Design	0.67	0.03	72.15%	−2.16%

**Table 11 medicina-60-02050-t011:** Statistical significances (*p*-values) for Figure 7.

Comparison	All Groups N = 61	Military N = 38	Civilian N = 21
MBO vs. TM	*p* = 0.0005	*p* = 0.0005	*p* = 0.03
MBSR vs. TM	*p* = 0.000000007	*p* = 0.00000004	*p* = 0.0006
OM vs. TM	*p* = 0.000013	*p* = 0.0005	*p* = 0.000007
MBSR vs. OM	*p* = 0.23	*p* = 0.09	*p* = 0.28
MBSR vs. MBO	*p* = 0.21	*p* = 0.90	*p* = 0.43
MBO vs. OM	*p* = 0.78	*p* = 0.32	*p* = 0.09

## Data Availability

All the original data analyses in this Systematic Review and Meta-Analysis is publicly available in the research papers cited, Summary data for each study are available in the Appendix A associated with this paper.

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
