# Peer review of "Effectiveness of Meditation Techniques in Treating Post-Traumatic Stress Disorder: A Systematic Review and Meta-Analysis"

_medicina, 2024, doi:10.3390/medicina60122050_

Round 1

Reviewer 1 Report

Comments and Suggestions for Authors

Esteemed Authors: David W. Orme-Johnson, Vernon A. Barnes, Brian Rees , Jean Tobin and Kenneth G. Walton

        Thank you for the chance to review your manuscript, "Efficacy of Meditation Strategies in the Treatment of Post-Traumatic Stress Disorder: Systematic Review and Meta-Analysis". So, first, I wish to commend you for doing such a good thing and synthesizing so much research to help us see how meditation might help with PTSD. The work in compiling 61 studies, carrying out powerful meta-analyses, and uncovering subtle trends across populations and methods is admirable.

          This is an excellent review to contribute to the research community as it explains how meditation modality’s compare with one another and highlights key directions for further research. The results on the transformative power of TM are especially potent and have come in line with increasing focus on non-pharmacological treatment for PTSD.

Objectives and rationale

        You’re following a specific and very specific set of goals: which meditation approaches are best for PTSD? But the "other meditations (OM)" inclusion doesn’t have specific rationale to explain why it was added. If it were possible to briefly explain why these techniques were selected and why they fit into the study’s general framework, it would strengthen this part.

Methodology
          That PRISMA protocol you follow and the method section you explain is admirable. But missing information (i.e., guessing average age) needs more explaining so that the readers can see how this affects outcomes. Additionally, putting all practices under the broad umbrella of "OM" would obscure valuable differences. A more detailed dissection in this group would be clearer and more precise.

Statistical analyses

           The statistical approach is appropriate and you are making good use of Hedges’s g, prediction intervals, and meta-regression. Yet ignoring outliers, like Congolese refugee research, is recognized without ever discussing what that means in the great perspective. A sensitivity analysis showing the effects of such exclusions would support your claim.

        Also, explained variance (I2 > 86%) suggests that there could be confounders affecting the outcomes. Talking about these potential causes in the manuscript would also contextualize the result and point future studies.

Results and interpretation

        The finding that TM is most effective is also well-supported and a great contribution to the literature. Having said that, the small effect sizes of MBSR and OM are still worth accounting for as they still stand as helpful treatments for PTSD. The findings by population (eg, veterans vs. civilians) are interesting, but unexplored. The deeper the discussion, the more interpretation we can offer about these distinctions.

Figures and tables

       To show the relationship between dropout or completion rate and intervention impact would give this data more practical meaning. Figures 2–5 could also use a little more explanation caption so they make sense.

Strengths
           You have a rigorous and methodologically robust study with 61 studies on diverse populations and forms of meditation. There is significant benefit in meta-regression, which provides an explanation for the effects of age and trauma type.

Limitations
            You have already acknowledged a number of limitations: low numbers, randomized meditation. But there are other concerns, including biases in study selection and the omniscient categorization of OM. It would be more transparent and credible if these points are discussed in the limitations section.

Structure and language

         The text is clear but the flow between sections might have been better: tying results to the conversation, for instance, would lead readers more easily through your story. The text is accessible but certain passages – especially the part about statistical approaches – are very hard for the common reader to read through. Eliminating these sections would render the study accessible but without sacrificing rigour.

Closing thoughts

         It is timely and significant to have you contribute to the literature about treatment for PTSD. You can make it more clear, balanced and impactful by implementing the suggestions above. You should carry on doing so because your results set the stage for meditation-based treatments for PTSD and better outcomes for patients.

       Thanks again for the chance to read your manuscript. I look forward to seeing how this work advances the scholarship and treatment of PTSD.

Yours truly,

Serving Peer Reviewer at Medicina,

Author Response

Reviews of Medicina 3342190

Dr. David W. Orme-Johnson’s comments and replies to reviewers’ points are in Blue. Changes that were made in the manuscript Medicina 3342190 are in blue italics. 

General Points

Please revise the manuscript found at the above link according to the reviewers' comments and upload the revised file within 5 days. Note the following check-list:

(I) Ensure all references are relevant to the content of the manuscript. Done

(II) Highlight any revisions to the manuscript, so editors and reviewers can see any changes made. Done

(III) Provide a cover letter to respond to the reviewers’ comments and explain, point by point, the details of the manuscript revisions. Done

(IV) If the reviewer(s) recommended references, critically analyze them to ensure that their inclusion would enhance your manuscript. If you believe these references are unnecessary, you should not include them. Done

(V) If you found it impossible to address certain comments in the review reports, include an explanation in your appeal. Done.

Reviewer 1.

Thank you for the chance to review your manuscript, "Efficacy of Meditation Strategies in the Treatment of Post-Traumatic Stress Disorder: Systematic Review and Meta-Analysis". So, first, I wish to commend you for doing such a good thing and synthesizing so much research to help us see how meditation might help with PTSD. The work in compiling 61 studies, carrying out powerful meta-analyses, and uncovering subtle trends across populations and methods is admirable.

Thank you for your appreciation. We hope it makes a useful contribution.
          This is an excellent review to contribute to the research community as it explains how meditation modality’s compare with one another and highlights key directions for further research. The results on the transformative power of TM are especially potent and have come in line with increasing focus on non-pharmacological treatment for PTSD.

Thank you.
Objectives and rationale
        You’re following a specific and very specific set of goals: which meditation approaches are best for PTSD? But the "other meditations (OM)" inclusion doesn’t have specific rationale to explain why it was added. If it were possible to briefly explain why these techniques were selected and why they fit into the study’s general framework, it would strengthen this part.

As part of filling out the Prisma check list, we added a new flow diagram of the literature search (Figure 1, p. 8) and more extended descriptions of the treatment categories, which includes OM.  We added OM techniques as a category because we wanted our study to include all meditation techniques used to treat PTSD. OM includes meditation techniques that are neither TM nor Mindfulness-based. In the new text on page 22 we state our empirical rationale for this category. 

“These 61 studies were then logically divided into four categories of meditation, including the two categories that are uniformly defined, Transcendental Meditation (TM) and Mindfulness-Based Stress Reduction (MBSR). The first of the other two categories is Mindfulness-Based Other (MBO), which are mindfulness-based techniques but not MBSR. MBO techniques are often derived from MBSR but do not follow the MBSR protocol. MBO techniques often experiment with different lengths of training, mode of presentation (such as via telecommunications), or with additional techniques, such as Hatha Yoga. The fourth category of meditation is Other Meditations (OM), which are neither TM nor MBSR nor MBO. Examples of OM include compassion meditation, Kriya yoga, Adopted Mantra Meditation, and Mantrum Repetition meditation.”

The OM category has a diversity of techniques, and Table 5 on page 12 displays stats for each study, Hedges’ g, SE, Variance, 95% CI, Z and p-value. This information may be useful for the reader to find a meaningful way subdivide the OM studies. On page 10 we list the kinds of trauma patients on which OM techniques have been used (e.g., military, sexual trauma, traumatized youth in residential care).

Table 2 on page 10 summarizes implementation stats on OM and the other categories of meditation, and page 11 shows the research quality scores.  Table 7 on page 14 shows the variance of g scores, and page 18 the non-significance of age for predicting g for OM. Figure 7 and Table 11 on page 23 show the effects and significance of OM for military and civilian populations compared to the other categories of meditation.

Because Mantrum Repetition (MR) is cited as promising in the multisite study of meditation by the military commissioned by the U.S. military, we included a subgroup analysis comparing it with the two most studies meditation techniques, TM and MBSR. See page 21, Subgroup Analysis of Mantrum Repetition (MR), MBSR, and TM in the U.S. We found that MR had an effect size of g = -0.69 compared to -0.41 for MBSR and -1.06 for TM. We found no evidence of publication bias for any of the treatments.

In the Discussion on page 24 point out that our meta-analysis found more studies and larger effects than previous systematic reviews on OM, MBSR, and MBO techniques. Larger effects were because we reported within-group effects whereas other meta-analyses reported between group effects. We found more studies because using within-group effects enabled us to include Single Group case studies, as well as CPs and RCTs, whereas other meditations only included RCTs.

On page 24 we also point out that the large effects of TM are in accord with previous reviews.

Methodology
          That PRISMA protocol you follow and the method section you explain is admirable. But missing information (i.e., guessing average age) needs more explaining so that the readers can see how this affects outcomes. Additionally, putting all practices under the broad umbrella of "OM" would obscure valuable differences. A more detailed dissection in this group would be clearer and more precise.

I think the reviewer is referring to this statement in the methodology.

  1. 11. We included all participants diagnosed with PTSD, all types of traumas, all age groups, all gender identifications, all ethnicities, all countries. When data were missing on age, we used the mean age of their meditation category. This was for the purpose of multiple meta-regression analysis, in which one study with missing data would reduce the N for all comparisons.

This was not “guessing average age”. It was using the average of the other studies in the treatment category to fill in missing data. There were very few cases where data on age were not available, and very few studies with young subjects, most were middle aged. This resulted in a data set with a truncated range that is not well suited to studying the effects of age. We found (page 24) that age was a significant covariate for TM and MBO, but not for OM and MBSR. We discuss on page 24: However, none of these studies were designed to study the effects of age, which we suggest will be important for future studies. We added: Studies on the effects of age would need to be designed to have a systematic spread of age groups treated under the same or similar conditions, the same populations using the same meditation technique.”

I checked the study characteristics of the OM category and found was study with young subjects (Schuurmans, 2020, mean age 14) and these subjects did not show usually reduced PTSD. But then, they were practicing a Game-Based Meditation, so the lack of effect could have been due to the technique being ineffective for reducing PTSD. Maybe an age effect would be found if they were practicing a more effective technique. The potential effects of age need to be studies further, and I am adding a comment in red above to the paper (blue in the paper).

Statistical analyses
           The statistical approach is appropriate and you are making good use of Hedges’s g, prediction intervals, and meta-regression. Yet ignoring outliers, like Congolese refugee research, is recognized without ever discussing what that means in the great perspective. A sensitivity analysis showing the effects of such exclusions would support your claim.

We reported the effects of the two studies on Congolese refugees whose were 3 ad 5 times larger than summary effect of all other studies (p. 7). We found that their removal decreased the overall summary effect of TM from -1.31 to -1.19, with a p-value that was essentially zero. The span of studies’ effects was reduced by 25%. I2 = 91.28% with the outliers included to 86.17% with the outliers excluded, which left considerable heterogeneity to explain, which we addressed with meta-regression.

In a second paper from this meta-analysis, which is on the between-group effects of TM studies, which we are also submitting to Medicina, we speculate on the reasons for this outlier.

The Second Congo War (1998-2003), also known as the Great African War, involved nine African countries and around twenty-five armed groups who killed 5.4 million people and forced an estimated 80,000 refugees to flee96. Rees and colleagues published two studies of the effects of TM on refugees from the Democratic Republic of the Congo. They were staying around Kampala, Uganda in temporary shelters, such as churches or rented accommodations. They were typically unemployed and had minimal access to mental health services. This population had been exposed to combat stress, sexual assault, torture, and/or forced to witness the abuse or killing of loved ones. …In a Zoom call with this population by one of the authors (DOJ), their urgent question was “can you help us get out of this situation”. Such large effects show the study is an outlier. Even so, the clinically meaningful effects it found are consistent with those of the other studies and call for larger scale studies investigating the efficacy of TM practice across cultures … The large changes in effect sizes may have been due to experimental-demand effects in the TM participants to please the teachers, in the hopes that they may help get them into a better situation. In a Zoom call with this population by one of the authors (DOJ), their urgent question was “can you help us get out of this situation”.

Also, explained variance (I2 > 86%) suggests that there could be confounders affecting the outcomes. Talking about these potential causes in the manuscript would also contextualize the result and point future studies.

This is a good point. I added this comment to the discussion on page 24.

Our meta-regression found that treatment group, age, trauma groups, research design and military vs. civilian study populations were significant covariates in the regression on effect size (Hedges’s g). Yet even our best multivariate meta-regression model of these variables only reduced the unexplained variance (I2) to 72%. There is still much to be learned about what factors can enhance the effectiveness meditation treatments and we hope that our study will inspire new avenues of research.

Differences among meditation techniques.

Results and interpretation
        The finding that TM is most effective is also well-supported and a great contribution to the literature. Having said that, the small effect sizes of MBSR and OM are still worth accounting for as they still stand as helpful treatments for PTSD. The findings by population (eg, veterans vs. civilians) are interesting, but unexplored. The deeper the discussion, the more interpretation we can offer about these distinctions.

In the introduction (p. 4), in the history of using meditation for treating PTSD, give a brief history of “mindfulness” in Western psychology, and we cited several meta-analyses MBSR and some OM for PTSD. These previous papers discuss these issues, and the literature is full of theoretical discussions of mindfulness. It’s a big field to cover, and our paper was already too long to go into it. Moreover, the Heffner, 2014 multi-cite study measured characteristics of mindfulness, and change in mindfulness scores didn’t seem to predict change in PTSD. Mindfulness doesn’t seem a useful concept with regards to PTSD. We quote Chiesa, A.; Malinowski, 2011, “the currently applied mindfulness-based interventions show large differences in the way mindfulness is conceptualized and practiced (Chiesa, A.; Malinowski, P., Mindfulness-based approaches: are they all the same? Journal of Clinical Psychology 2011, 67 (4), 404-424.”

My personal view is that the prevalent understanding of mindfulness among psychologists is a misunderstanding of what Budda meant, turning Budda’s description of a state into a practice. Budda’s disciples could feel that he was different from them and when they asked him what he was experiencing he said equanimity, in the present moment, bliss. In our time that has become interpreted to mean that you practice open monitoring without judgement (equanimity), focused attention (in the present moment), mood making (bliss). But these outcomes, not practices. The practice is to transcend, by allowing the mind to settle inward to transcendental consciousness. In the Discussion (p. 25) we have documented that the physiological correlates of transcending are in the opposite direction of stress and that this is why TM is more effective in treating PTSD than all other meditation techniques which take the mind in activity of focusing on something, or monitoring something. These meditation activities will not allow the mind to transcend and hence the meditator will not get the benefits of transcending. A more rested mind and nervous system from transcending is what gives rise to being more in the moment, happier and more blissful, not trying to practice these outcomes.

We say a little about this in the proposed mechanism of TM in the Discussion on p. 25. The first physiological research on TM in the early 1970’s attests to its effortlessness. It found that TM produces effects that are opposite to the fight-or-flight stress response1-4. This physiological state came to be referred to as a restfully alert, hypometabolic, physiologic state2 or “restful alertness”. Whereas PTSD can be associated with increased heart rate, rapid breathing, muscle tension, pupil dilation, or increased blood pressure, effects of TM tend to be the opposite. A meta-analysis of 32 studies found that, during TM, respiratory rate, skin conductance, and plasma lactate level significantly decrease compared to controls resting with eyes closed5.

We go on to review the evidence showing the difference among meditation techniques on variable associated with transcending.

My next project after this meta-analysis is to write a book about my interactions with Maharishi since 1970 re expressing the knowledge of enlightenment from ancient India in term of modern scientific research and theory.

Thank you for giving us an opportunity to talk/write more about it.

Figures and tables

       To show the relationship between dropout or completion rate and intervention impact would give this data more practical meaning. Figures 2–5 could also use a little more explanation caption so they make sense.

Thank you. It would be very valuable to know why some people drop out and don’t complete meditation treatments to learn how to make the treatments more effective. However, we found in coding the studies that data on dropout and completion were not uniformly reported across studies, sometimes sparsely reported, if at all. We do not feel we had a good enough data to enable quantitative exploration of relationships with outcomes.

We did add these suggests in the section on Future Research, p.27.

In coding these studies, we found that implementation statistics on dropouts, completion rates, and regularity of meditation practices were not reliably reported across studies. This makes it difficult to quantitatively study the effects of these variables on treatment outcomes. We suggest that future research make sure to include these measures, which may be useful in finding ways to improve compliance and effectiveness of meditation in the treatment of PTSD.

We suggest adding measures of transcending to all studies on meditation. We have made a case that transcending is the key to the effectiveness of TM. We suggest that bio markers of transcending, such as slowing of breath, reduce spontaneous skin resistance responses, and alpha1 (7-10 Hz) EEG power and coherence, could be measured along with any meditation practice to see if they produce a degree of transcending and whether this is predictive of efficacy in reducing symptoms of trauma. Are there other meditation techniques, or different forms of known ones, that enable transcending? There has been much discussion and research on mindfulness as the key ingredient of successful meditation. Now we are suggesting a different paradigm of study transcending as potentially the key ingredient. In this meta-analysis, we have argued that transcending is the reason that TM is successful and not transcending is the reason that other meditation techniques are of limited efficacy. Perhaps a systematic search for transcending will lead to discovery of other meditation techniques that result in it.”

Strengths
           You have a rigorous and methodologically robust study with 61 studies on diverse populations and forms of meditation. There is significant benefit in meta-regression, which provides an explanation for the effects of age and trauma type.

Thank you. Always more to do!

Limitations
            You have already acknowledged a number of limitations: low numbers, randomized meditation. But there are other concerns, including biases in study selection and the omniscient categorization of OM. It would be more transparent and credible if these points are discussed in the limitations section. This is a delicate issue, because everyone has their world view and its associated biases. We used scientific methodologies designed to minimize the effects of bias. To address the issue of potential bias, we added this text to the paper in the section Limitations, pp. 27, 28.

“Everyone has a world view born of their life experiences, which potentially could be a source of bias in scientific research. We used funnel plot analyses, which found no evidence of publication bias for any category of meditation. We included all the studies we could locate, which was more studies than previous meta-analyses. This is because our selection criteria were more inclusive and our study was up to date, including many recent studies that had not been reviewed before. To be transparent we presented the data for each individual study in Tables 3-6 and in the Appendices. The summary effects for MBSR, MBO, OM and TM were similar in magnitude to those reported in previous systematic reviews by researchers independent of TM or MIU giving confidence that our results were not biased towards over reporting TM nor underreporting other techniques. Furthermore, the effects for studies conducted at institutions independent of MIU of TM practice were of similar magnitude to the effects of studies that are associated with TM and MIU, as has been found in other meta-analysis on psychological effects of meditation6.”

Structure and language

         The text is clear but the flow between sections might have been better: tying results to the conversation, for instance, would lead readers more easily through your story. The text is accessible but certain passages – especially the part about statistical approaches – are very hard for the common reader to read through. Eliminating these sections would render the study accessible but without sacrificing rigour.

We appreciate your point that the paper is difficult to read in places. We made a point to define all the statistical term, but they are still technical. They are necessarily technical to provide a full understanding of the methods used. The article is not intended for the common reader. We aim to present rigorous scientific research, requiring detailed explanations of study methodologies, statistical analyses, and specific terminology to ensure transparency and allow other researchers to replicate the findings, all within the strict standards of peer review which demands a high level of precision and clarity in reporting data and results.

That being said, we went over the paper and tried to improve the flow and comprehensibility.

On pp. 8, 9 we added a new flow chart according to Prisma 2020 and descriptions that makes it clearer how we conducted out systematic search.

  1. 9-11 we present tables that summarize study characteristics, implementation statistics, and research quality. We put with tables with detail for each individual study in as Appendices. This will be a little award to navigate for readers who want to see that level while reading the main text , but we found that putting all the big tables in the Appendices into the text made it unmanageable to read, so we put them in the Appendices.
  2. 11-13, we put the full Tables 3-6 of basic outcome statistics on each individual study (effect sizes Hedges’s g, SE, variance, etc.) in the text because these are the main result of interest.
  3. 14, Table 8: Prediction interval is perhaps less well understood statistic, so we put in an explanation of it on p. 15. “In the lower section of Table 8, under Mixed effects model, we see the prediction intervals, which are the indicators of heterogeneity. The prediction interval (PI) is the range of effect sizes that future studies on all comparable populations would likely fall into (if we assume that the true effects are normally distributed in g units). We added this note to this section on p 15.

“It is a common error in meta-analyses to believe that I2 is the measure of heterogeneity. It is not. I2 estimates the proportion of unexplained variance. PI estimates the spread of effects of studies, which is the heterogeneity. It is an important clarify what heterogeneity is in systematic reviews because the PI may predict that some treatments are likely to have adverse effects, something that clinicians need to be aware of,  that is not provided by the I2 statistic7, 8.”

Closing thoughts

         It is timely and significant to have you contribute to the literature about treatment for PTSD. You can make it more clear, balanced and impactful by implementing the suggestions above. You should carry on doing so because your results set the stage for meditation-based treatments for PTSD and better outcomes for patients.

Thank you for your constructive comments. I think replying to them has made our paper stronger.
       Thanks again for the chance to read your manuscript. I look forward to seeing how this work advances the scholarship and treatment of PTSD.

Thank you for taking the time to review it and make so many constructive suggestions!

Yours truly,

Serving Peer Reviewer at Medicina,

Reviewer 2 Report

Comments and Suggestions for Authors

It is a valuable study that meta-analyzed and systematically reviewed studies that verified the effectiveness of meditation in treating patients with post-traumatic stress disorder. In particular, the meta-regression provided clinically useful information. Your analysis of the differences between Civilian and Military also seems very clinically useful. Therefore, I think it would be a high-quality manuscript if only a few things were supplemented. There are some things that could be improved:

1. The information presented in the introduction under the subheadings 'The Problem', 'Prevalence', and 'Treatment' contains very general information. Therefore, please delete the subheadings and present a persuasive argument that PTSD is serious and its prevalence is not negligible, and what methods have been applied to treat it. Then, it would be good to describe “Meditation in the treatment of PTSD”.

2. It's not that you haven't presented them, but it would be nice if you presented them in a more organized manner.

3. If you could present the RoB table in a more presentable way, I think readers would get more useful information.

4. It would be helpful to readers if you could summarize the clinical implications of the findings of this study in more detail.

Author Response

Reviewer #2

Open Review

Quality of English Language

(x) The quality of English does not limit my understanding of the research.
( ) The English could be improved to more clearly express the research.

Yes

Can be improved

Must be improved

Not applicable

Does the introduction provide sufficient background and include all relevant references?

( )

(x)

( )

( )

Is the research design appropriate?

(x)

( )

( )

( )

Are the methods adequately described?

(x)

( )

( )

( )

Are the results clearly presented?

(x)

( )

( )

( )

Are the conclusions supported by the results?

( )

(x)

( )

( )

Comments and Suggestions for Authors

It is a valuable study that meta-analyzed and systematically reviewed studies that verified the effectiveness of meditation in treating patients with post-traumatic stress disorder. In particular, the meta-regression provided clinically useful information. Your analysis of the differences between Civilian and Military also seems very clinically useful. Therefore, I think it would be a high-quality manuscript if only a few things were supplemented. There are some things that could be improved:

  1. The information presented in the introduction under the subheadings 'The Problem', 'Prevalence', and 'Treatment' contains very general information. Therefore, please delete the subheadings and present a persuasive argument that PTSD is serious and its prevalence is not negligible, and what methods have been applied to treat it. Then, it would be good to describe “Meditation in the treatment of PTSD”.

We have gone over the Introduction and added some phrases that address this issue of the seriousness and prevalence of PTSD. We indicate the magnitude of the problem by quoting the WHO estimates that 320 million people will have PTSD, as well as providing stats on the seriousness in the U.S. military.

  1. It's not that you haven't presented them, but it would be nice if you presented them in a more organized manner.

Thank you for this suggestion, but on reviewing the text we feel the headings 'The Problem', 'Prevalence', and 'Treatment' are a logical progression and do help with the organization. We decided to leave in.

  1. If you could present the RoB table in a more presentable way, I think readers would get more useful information.

What we did was present the risk of bias before presenting the main results. Our logic was to establish no risk of bias to give confidence in the results that were forthcoming.

Here is how we presented risk of bias. In the Methods section (p. 7) we present our method of assessing risk of bias. On p. 11 we present Appendix B, which defines the bias analysis tool we used (CLEAR scores), and Appendix C , which presents tables with CLEAR scores for each study, as well as t-tests and graphs that show that the strengths and weaknesses of research quality were similar and not statistical different for MBO, MBSR, OM, and TM.

On p. 11, to make the point clearer, we added a concluding sentence saying that “The lack of differences between types of meditation on research quality indicates that it is unlikely that research bias could cause differences in their effectiveness.”

On p. 22 we present the results of no evidence of publication bias for the subgroup analysis of MR, MBSR, or TM in the military. We refer the reader to details of the analysis in Supplementary Materials, Appendix F: Military and Civilian Comparisons, Part 1.

In the Discussion (pp. 27, 28) we added new text that considers the issue of bias more generally and the tools we used to reduce it. E.g., “Everyone has a world view born of their life experiences, which potentially could be a source of bias in scientific research. We used funnel plot analyses, which found no evidence of publication bias for any category of meditation. We included all the studies we could locate, which was more studies than previous meta-analyses. This is because our selection criteria were more inclusive, and our study was up to date. We included many recent studies that had not been reviewed before….”

  1. It would be helpful to readers if you could summarize the clinical implications of the findings of this study in more detail. Thank you. We added this sentence to the end of the Conclusions, p. 28.

“This research suggests that the wide implementation of this very effective treatment for PTSD would have significant cost savings and  clinical implications, including: reduced symptom severity, improved quality of life, increased functional ability, decreased reliance on medication, better social relationships, and potentially a greater capacity for resilience in individuals experiencing post-traumatic stress disorder; essentially allowing people with PTSD to manage their symptoms and participate more fully in daily life.”

Submission Date

13 November 2024

Date of this review

24 Nov 2024 07:05:20

  1. Wallace, R. K., Physiological effects of Transcendental Meditation. Science 1970, 167, 1751–1754.
  2. Wallace, R. K.; Benson, H.; Wilson, A. F., A wakeful hypometabolic physiologic state. American Journal of Physiology 1971, 221, 795-799.
  3. Wallace, R. K.; et, a., The Physiology of Meditation. Scientific American 1972, 226, 84-90.
  4. Orme-Johnson, D. W., Autonomic stability and Transcendental Meditation. Psychosomatic Medicine 1973, 35, 341-349.
  5. Dillbeck, M. C.; Orme-Johnson, D. W., Physiological differences between Transcendental Meditation and rest. American Psychologist 1987, 42, 879–881.
  6. Orme-Johnson, D. W.; Dillbeck, M. C., Methodological Concerns for Meta-Analyses of Meditation: Comment on Sedlmeier et al. (2012). Psychological Bulletin 2014, 140 (2), 610–616.
  7. Borenstein, M., Research Note: In a meta-analysis, the I2 index does not tell us how much the effect size varies across studies. Journal of Physiotherapy 2020, 66 (2), 135-139.
  8. Borenstein, M.; Higgins, J. P.;  Hedges, L. V.; Rothstein, H. R., Basics of meta-analysis: I2 is not an absolute measure of heterogeneity. Research Synthesis Methods 2017, 8 (1), 177-188.

    Reviewer #2

    Open Review

    Quality of English Language

    (x) The quality of English does not limit my understanding of the research.
    ( ) The English could be improved to more clearly express the research.

    Yes

    Can be improved

    Must be improved

    Not applicable

    Does the introduction provide sufficient background and include all relevant references?

    ( )

    (x)

    ( )

    ( )

    Is the research design appropriate?

    (x)

    ( )

    ( )

    ( )

    Are the methods adequately described?

    (x)

    ( )

    ( )

    ( )

    Are the results clearly presented?

    (x)

    ( )

    ( )

    ( )

    Are the conclusions supported by the results?

    ( )

    (x)

    ( )

    ( )

    Comments and Suggestions for Authors

    It is a valuable study that meta-analyzed and systematically reviewed studies that verified the effectiveness of meditation in treating patients with post-traumatic stress disorder. In particular, the meta-regression provided clinically useful information. Your analysis of the differences between Civilian and Military also seems very clinically useful. Therefore, I think it would be a high-quality manuscript if only a few things were supplemented. There are some things that could be improved:

    1. The information presented in the introduction under the subheadings 'The Problem', 'Prevalence', and 'Treatment' contains very general information. Therefore, please delete the subheadings and present a persuasive argument that PTSD is serious and its prevalence is not negligible, and what methods have been applied to treat it. Then, it would be good to describe “Meditation in the treatment of PTSD”.

    We have gone over the Introduction and added some phrases that address this issue of the seriousness and prevalence of PTSD. We indicate the magnitude of the problem by quoting the WHO estimates that 320 million people will have PTSD, as well as providing stats on the seriousness in the U.S. military.

    1. It's not that you haven't presented them, but it would be nice if you presented them in a more organized manner.

    Thank you for this suggestion, but on reviewing the text we feel the headings 'The Problem', 'Prevalence', and 'Treatment' are a logical progression and do help with the organization. We decided to leave in.

    1. If you could present the RoB table in a more presentable way, I think readers would get more useful information.

    What we did was present the risk of bias before presenting the main results. Our logic was to establish no risk of bias to give confidence in the results that were forthcoming.

    Here is how we presented risk of bias. In the Methods section (p. 7) we present our method of assessing risk of bias. On p. 11 we present Appendix B, which defines the bias analysis tool we used (CLEAR scores), and Appendix C , which presents tables with CLEAR scores for each study, as well as t-tests and graphs that show that the strengths and weaknesses of research quality were similar and not statistical different for MBO, MBSR, OM, and TM.

    On p. 11, to make the point clearer, we added a concluding sentence saying that “The lack of differences between types of meditation on research quality indicates that it is unlikely that research bias could cause differences in their effectiveness.”

    On p. 22 we present the results of no evidence of publication bias for the subgroup analysis of MR, MBSR, or TM in the military. We refer the reader to details of the analysis in Supplementary Materials, Appendix F: Military and Civilian Comparisons, Part 1.

    In the Discussion (pp. 27, 28) we added new text that considers the issue of bias more generally and the tools we used to reduce it. E.g., “Everyone has a world view born of their life experiences, which potentially could be a source of bias in scientific research. We used funnel plot analyses, which found no evidence of publication bias for any category of meditation. We included all the studies we could locate, which was more studies than previous meta-analyses. This is because our selection criteria were more inclusive, and our study was up to date. We included many recent studies that had not been reviewed before….”

    1. It would be helpful to readers if you could summarize the clinical implications of the findings of this study in more detail. Thank you. We added this sentence to the end of the Conclusions, p. 28.

    “This research suggests that the wide implementation of this very effective treatment for PTSD would have significant cost savings and  clinical implications, including: reduced symptom severity, improved quality of life, increased functional ability, decreased reliance on medication, better social relationships, and potentially a greater capacity for resilience in individuals experiencing post-traumatic stress disorder; essentially allowing people with PTSD to manage their symptoms and participate more fully in daily life.”

    Submission Date

    13 November 2024

    Date of this review

    24 Nov 2024 07:05:20

    1. Wallace, R. K., Physiological effects of Transcendental Meditation. Science 1970, 167, 1751–1754.
    2. Wallace, R. K.; Benson, H.; Wilson, A. F., A wakeful hypometabolic physiologic state. American Journal of Physiology 1971, 221, 795-799.
    3. Wallace, R. K.; et, a., The Physiology of Meditation. Scientific American 1972, 226, 84-90.
    4. Orme-Johnson, D. W., Autonomic stability and Transcendental Meditation. Psychosomatic Medicine 1973, 35, 341-349.
    5. Dillbeck, M. C.; Orme-Johnson, D. W., Physiological differences between Transcendental Meditation and rest. American Psychologist 1987, 42, 879–881.
    6. Orme-Johnson, D. W.; Dillbeck, M. C., Methodological Concerns for Meta-Analyses of Meditation: Comment on Sedlmeier et al. (2012). Psychological Bulletin 2014, 140 (2), 610–616.
    7. Borenstein, M., Research Note: In a meta-analysis, the I2 index does not tell us how much the effect size varies across studies. Journal of Physiotherapy 2020, 66 (2), 135-139.
    8. Borenstein, M.; Higgins, J. P.;  Hedges, L. V.; Rothstein, H. R., Basics of meta-analysis: I2 is not an absolute measure of heterogeneity. Research Synthesis Methods 2017, 8 (1), 177-188.

      Reviewer #2

      Open Review

      Quality of English Language

      (x) The quality of English does not limit my understanding of the research.
      ( ) The English could be improved to more clearly express the research.

      Yes

      Can be improved

      Must be improved

      Not applicable

      Does the introduction provide sufficient background and include all relevant references?

      ( )

      (x)

      ( )

      ( )

      Is the research design appropriate?

      (x)

      ( )

      ( )

      ( )

      Are the methods adequately described?

      (x)

      ( )

      ( )

      ( )

      Are the results clearly presented?

      (x)

      ( )

      ( )

      ( )

      Are the conclusions supported by the results?

      ( )

      (x)

      ( )

      ( )

      Comments and Suggestions for Authors

      It is a valuable study that meta-analyzed and systematically reviewed studies that verified the effectiveness of meditation in treating patients with post-traumatic stress disorder. In particular, the meta-regression provided clinically useful information. Your analysis of the differences between Civilian and Military also seems very clinically useful. Therefore, I think it would be a high-quality manuscript if only a few things were supplemented. There are some things that could be improved:

      1. The information presented in the introduction under the subheadings 'The Problem', 'Prevalence', and 'Treatment' contains very general information. Therefore, please delete the subheadings and present a persuasive argument that PTSD is serious and its prevalence is not negligible, and what methods have been applied to treat it. Then, it would be good to describe “Meditation in the treatment of PTSD”.

      We have gone over the Introduction and added some phrases that address this issue of the seriousness and prevalence of PTSD. We indicate the magnitude of the problem by quoting the WHO estimates that 320 million people will have PTSD, as well as providing stats on the seriousness in the U.S. military.

      1. It's not that you haven't presented them, but it would be nice if you presented them in a more organized manner.

      Thank you for this suggestion, but on reviewing the text we feel the headings 'The Problem', 'Prevalence', and 'Treatment' are a logical progression and do help with the organization. We decided to leave in.

      1. If you could present the RoB table in a more presentable way, I think readers would get more useful information.

      What we did was present the risk of bias before presenting the main results. Our logic was to establish no risk of bias to give confidence in the results that were forthcoming.

      Here is how we presented risk of bias. In the Methods section (p. 7) we present our method of assessing risk of bias. On p. 11 we present Appendix B, which defines the bias analysis tool we used (CLEAR scores), and Appendix C , which presents tables with CLEAR scores for each study, as well as t-tests and graphs that show that the strengths and weaknesses of research quality were similar and not statistical different for MBO, MBSR, OM, and TM.

      On p. 11, to make the point clearer, we added a concluding sentence saying that “The lack of differences between types of meditation on research quality indicates that it is unlikely that research bias could cause differences in their effectiveness.”

      On p. 22 we present the results of no evidence of publication bias for the subgroup analysis of MR, MBSR, or TM in the military. We refer the reader to details of the analysis in Supplementary Materials, Appendix F: Military and Civilian Comparisons, Part 1.

      In the Discussion (pp. 27, 28) we added new text that considers the issue of bias more generally and the tools we used to reduce it. E.g., “Everyone has a world view born of their life experiences, which potentially could be a source of bias in scientific research. We used funnel plot analyses, which found no evidence of publication bias for any category of meditation. We included all the studies we could locate, which was more studies than previous meta-analyses. This is because our selection criteria were more inclusive, and our study was up to date. We included many recent studies that had not been reviewed before….”

      1. It would be helpful to readers if you could summarize the clinical implications of the findings of this study in more detail. Thank you. We added this sentence to the end of the Conclusions, p. 28.

      “This research suggests that the wide implementation of this very effective treatment for PTSD would have significant cost savings and  clinical implications, including: reduced symptom severity, improved quality of life, increased functional ability, decreased reliance on medication, better social relationships, and potentially a greater capacity for resilience in individuals experiencing post-traumatic stress disorder; essentially allowing people with PTSD to manage their symptoms and participate more fully in daily life.”

      Submission Date

      13 November 2024

      Date of this review

      24 Nov 2024 07:05:20

      1. Wallace, R. K., Physiological effects of Transcendental Meditation. Science 1970, 167, 1751–1754.
      2. Wallace, R. K.; Benson, H.; Wilson, A. F., A wakeful hypometabolic physiologic state. American Journal of Physiology 1971, 221, 795-799.
      3. Wallace, R. K.; et, a., The Physiology of Meditation. Scientific American 1972, 226, 84-90.
      4. Orme-Johnson, D. W., Autonomic stability and Transcendental Meditation. Psychosomatic Medicine 1973, 35, 341-349.
      5. Dillbeck, M. C.; Orme-Johnson, D. W., Physiological differences between Transcendental Meditation and rest. American Psychologist 1987, 42, 879–881.
      6. Orme-Johnson, D. W.; Dillbeck, M. C., Methodological Concerns for Meta-Analyses of Meditation: Comment on Sedlmeier et al. (2012). Psychological Bulletin 2014, 140 (2), 610–616.
      7. Borenstein, M., Research Note: In a meta-analysis, the I2 index does not tell us how much the effect size varies across studies. Journal of Physiotherapy 2020, 66 (2), 135-139.
      8. Borenstein, M.; Higgins, J. P.;  Hedges, L. V.; Rothstein, H. R., Basics of meta-analysis: I2 is not an absolute measure of heterogeneity. Research Synthesis Methods 2017, 8 (1), 177-188.

        Reviewer #2

        Open Review

        Quality of English Language

        (x) The quality of English does not limit my understanding of the research.
        ( ) The English could be improved to more clearly express the research.

        Yes

        Can be improved

        Must be improved

        Not applicable

        Does the introduction provide sufficient background and include all relevant references?

        ( )

        (x)

        ( )

        ( )

        Is the research design appropriate?

        (x)

        ( )

        ( )

        ( )

        Are the methods adequately described?

        (x)

        ( )

        ( )

        ( )

        Are the results clearly presented?

        (x)

        ( )

        ( )

        ( )

        Are the conclusions supported by the results?

        ( )

        (x)

        ( )

        ( )

        Comments and Suggestions for Authors

        It is a valuable study that meta-analyzed and systematically reviewed studies that verified the effectiveness of meditation in treating patients with post-traumatic stress disorder. In particular, the meta-regression provided clinically useful information. Your analysis of the differences between Civilian and Military also seems very clinically useful. Therefore, I think it would be a high-quality manuscript if only a few things were supplemented. There are some things that could be improved:

        1. The information presented in the introduction under the subheadings 'The Problem', 'Prevalence', and 'Treatment' contains very general information. Therefore, please delete the subheadings and present a persuasive argument that PTSD is serious and its prevalence is not negligible, and what methods have been applied to treat it. Then, it would be good to describe “Meditation in the treatment of PTSD”.

        We have gone over the Introduction and added some phrases that address this issue of the seriousness and prevalence of PTSD. We indicate the magnitude of the problem by quoting the WHO estimates that 320 million people will have PTSD, as well as providing stats on the seriousness in the U.S. military.

        1. It's not that you haven't presented them, but it would be nice if you presented them in a more organized manner.

        Thank you for this suggestion, but on reviewing the text we feel the headings 'The Problem', 'Prevalence', and 'Treatment' are a logical progression and do help with the organization. We decided to leave in.

        1. If you could present the RoB table in a more presentable way, I think readers would get more useful information.

        What we did was present the risk of bias before presenting the main results. Our logic was to establish no risk of bias to give confidence in the results that were forthcoming.

        Here is how we presented risk of bias. In the Methods section (p. 7) we present our method of assessing risk of bias. On p. 11 we present Appendix B, which defines the bias analysis tool we used (CLEAR scores), and Appendix C , which presents tables with CLEAR scores for each study, as well as t-tests and graphs that show that the strengths and weaknesses of research quality were similar and not statistical different for MBO, MBSR, OM, and TM.

        On p. 11, to make the point clearer, we added a concluding sentence saying that “The lack of differences between types of meditation on research quality indicates that it is unlikely that research bias could cause differences in their effectiveness.”

        On p. 22 we present the results of no evidence of publication bias for the subgroup analysis of MR, MBSR, or TM in the military. We refer the reader to details of the analysis in Supplementary Materials, Appendix F: Military and Civilian Comparisons, Part 1.

        In the Discussion (pp. 27, 28) we added new text that considers the issue of bias more generally and the tools we used to reduce it. E.g., “Everyone has a world view born of their life experiences, which potentially could be a source of bias in scientific research. We used funnel plot analyses, which found no evidence of publication bias for any category of meditation. We included all the studies we could locate, which was more studies than previous meta-analyses. This is because our selection criteria were more inclusive, and our study was up to date. We included many recent studies that had not been reviewed before….”

        1. It would be helpful to readers if you could summarize the clinical implications of the findings of this study in more detail. Thank you. We added this sentence to the end of the Conclusions, p. 28.

        “This research suggests that the wide implementation of this very effective treatment for PTSD would have significant cost savings and  clinical implications, including: reduced symptom severity, improved quality of life, increased functional ability, decreased reliance on medication, better social relationships, and potentially a greater capacity for resilience in individuals experiencing post-traumatic stress disorder; essentially allowing people with PTSD to manage their symptoms and participate more fully in daily life.”

        Submission Date

        13 November 2024

        Date of this review

        24 Nov 2024 07:05:20

        1. Wallace, R. K., Physiological effects of Transcendental Meditation. Science 1970, 167, 1751–1754.
        2. Wallace, R. K.; Benson, H.; Wilson, A. F., A wakeful hypometabolic physiologic state. American Journal of Physiology 1971, 221, 795-799.
        3. Wallace, R. K.; et, a., The Physiology of Meditation. Scientific American 1972, 226, 84-90.
        4. Orme-Johnson, D. W., Autonomic stability and Transcendental Meditation. Psychosomatic Medicine 1973, 35, 341-349.
        5. Dillbeck, M. C.; Orme-Johnson, D. W., Physiological differences between Transcendental Meditation and rest. American Psychologist 1987, 42, 879–881.
        6. Orme-Johnson, D. W.; Dillbeck, M. C., Methodological Concerns for Meta-Analyses of Meditation: Comment on Sedlmeier et al. (2012). Psychological Bulletin 2014, 140 (2), 610–616.
        7. Borenstein, M., Research Note: In a meta-analysis, the I2 index does not tell us how much the effect size varies across studies. Journal of Physiotherapy 2020, 66 (2), 135-139.
        8. Borenstein, M.; Higgins, J. P.;  Hedges, L. V.; Rothstein, H. R., Basics of meta-analysis: I2 is not an absolute measure of heterogeneity. Research Synthesis Methods 2017, 8 (1), 177-188.

          Reviewer #2

          Open Review

          Quality of English Language

          (x) The quality of English does not limit my understanding of the research.
          ( ) The English could be improved to more clearly express the research.

          Yes

          Can be improved

          Must be improved

          Not applicable

          Does the introduction provide sufficient background and include all relevant references?

          ( )

          (x)

          ( )

          ( )

          Is the research design appropriate?

          (x)

          ( )

          ( )

          ( )

          Are the methods adequately described?

          (x)

          ( )

          ( )

          ( )

          Are the results clearly presented?

          (x)

          ( )

          ( )

          ( )

          Are the conclusions supported by the results?

          ( )

          (x)

          ( )

          ( )

          Comments and Suggestions for Authors

          It is a valuable study that meta-analyzed and systematically reviewed studies that verified the effectiveness of meditation in treating patients with post-traumatic stress disorder. In particular, the meta-regression provided clinically useful information. Your analysis of the differences between Civilian and Military also seems very clinically useful. Therefore, I think it would be a high-quality manuscript if only a few things were supplemented. There are some things that could be improved:

          1. The information presented in the introduction under the subheadings 'The Problem', 'Prevalence', and 'Treatment' contains very general information. Therefore, please delete the subheadings and present a persuasive argument that PTSD is serious and its prevalence is not negligible, and what methods have been applied to treat it. Then, it would be good to describe “Meditation in the treatment of PTSD”.

          We have gone over the Introduction and added some phrases that address this issue of the seriousness and prevalence of PTSD. We indicate the magnitude of the problem by quoting the WHO estimates that 320 million people will have PTSD, as well as providing stats on the seriousness in the U.S. military.

          1. It's not that you haven't presented them, but it would be nice if you presented them in a more organized manner.

          Thank you for this suggestion, but on reviewing the text we feel the headings 'The Problem', 'Prevalence', and 'Treatment' are a logical progression and do help with the organization. We decided to leave in.

          1. If you could present the RoB table in a more presentable way, I think readers would get more useful information.

          What we did was present the risk of bias before presenting the main results. Our logic was to establish no risk of bias to give confidence in the results that were forthcoming.

          Here is how we presented risk of bias. In the Methods section (p. 7) we present our method of assessing risk of bias. On p. 11 we present Appendix B, which defines the bias analysis tool we used (CLEAR scores), and Appendix C , which presents tables with CLEAR scores for each study, as well as t-tests and graphs that show that the strengths and weaknesses of research quality were similar and not statistical different for MBO, MBSR, OM, and TM.

          On p. 11, to make the point clearer, we added a concluding sentence saying that “The lack of differences between types of meditation on research quality indicates that it is unlikely that research bias could cause differences in their effectiveness.”

          On p. 22 we present the results of no evidence of publication bias for the subgroup analysis of MR, MBSR, or TM in the military. We refer the reader to details of the analysis in Supplementary Materials, Appendix F: Military and Civilian Comparisons, Part 1.

          In the Discussion (pp. 27, 28) we added new text that considers the issue of bias more generally and the tools we used to reduce it. E.g., “Everyone has a world view born of their life experiences, which potentially could be a source of bias in scientific research. We used funnel plot analyses, which found no evidence of publication bias for any category of meditation. We included all the studies we could locate, which was more studies than previous meta-analyses. This is because our selection criteria were more inclusive, and our study was up to date. We included many recent studies that had not been reviewed before….”

          1. It would be helpful to readers if you could summarize the clinical implications of the findings of this study in more detail. 

          Thank you. We added this sentence to the end of the Conclusions, p. 28.

          “This research suggests that the wide implementation of this very effective treatment for PTSD would have significant cost savings and  clinical implications, including: reduced symptom severity, improved quality of life, increased functional ability, decreased reliance on medication, better social relationships, and potentially a greater capacity for resilience in individuals experiencing post-traumatic stress disorder; essentially allowing people with PTSD to manage their symptoms and participate more fully in daily life.”

          Submission Date

          13 November 2024

          Date of this review

          24 Nov 2024 07:05:20

          These references are mostly for the reply to reviewer #1.

          1. Wallace, R. K., Physiological effects of Transcendental Meditation. Science 1970, 167, 1751–1754.
          2. Wallace, R. K.; Benson, H.; Wilson, A. F., A wakeful hypometabolic physiologic state. American Journal of Physiology 1971, 221, 795-799.
          3. Wallace, R. K.; et, a., The Physiology of Meditation. Scientific American 1972, 226, 84-90.
          4. Orme-Johnson, D. W., Autonomic stability and Transcendental Meditation. Psychosomatic Medicine 1973, 35, 341-349.
          5. Dillbeck, M. C.; Orme-Johnson, D. W., Physiological differences between Transcendental Meditation and rest. American Psychologist 1987, 42, 879–881.
          6. Orme-Johnson, D. W.; Dillbeck, M. C., Methodological Concerns for Meta-Analyses of Meditation: Comment on Sedlmeier et al. (2012). Psychological Bulletin 2014, 140 (2), 610–616.
          7. Borenstein, M., Research Note: In a meta-analysis, the I2 index does not tell us how much the effect size varies across studies. Journal of Physiotherapy 2020, 66 (2), 135-139.
          8. Borenstein, M.; Higgins, J. P.;  Hedges, L. V.; Rothstein, H. R., Basics of meta-analysis: I2 is not an absolute measure of heterogeneity. Research Synthesis Methods 2017, 8 (1), 177-188.
